# THINK BEFORE YOU ACT: DECISION TRANSFORMERS WITH INTERNAL MEMORY

## ABSTRACT

Decision transformer model-based decision-making agents have shown the ability to generalize across multiple tasks. However, their performance relies on massive data and computation. We argue that this inefficiency stems from the forgetting phenomenon, in which a model memorizes its behaviors in parameters throughout training. As a result, training on a new task may deteriorate the model's performance on previous tasks. In contrast to LLMs' implicit memory mechanism, the human brain utilizes distributed memory storage, which helps manage and organize multiple skills efficiently, mitigating the forgetting phenomenon. Thus inspired, we propose an internal memory module to store, blend, and retrieve information for different downstream tasks. Evaluation results show that the proposed method improves training efficiency and generalization in both Atari games and meta-world object manipulation tasks. Moreover, we demonstrate that memory fine-tuning further enhances the adaptability of the proposed architecture.

## 1 INTRODUCTION

Recently, with the tremendous success of decoder-only transformer models (Brown et al., 2020; OpenAI, 2023; Dosovitskiy et al., 2021; Touvron et al., 2023), an increasing number of researchers have focused on decoder-only transformer-based decision-making agents. As shown with GPT-3 (Brown et al., 2020) and follow-up work Kaplan et al. (2020); Clark et al. (2022), the generalization of these LLMs depends significantly on the model size, *i.e.* the number of parameters. This is partly because neural network parameters act as implicit memory (Neyshabur et al., 2019), enabling models to "memorize" a huge amount of training data by fitting these parameters. However, relying purely on scale has practical and ethical limits: there are economic and ecological costs, it re-

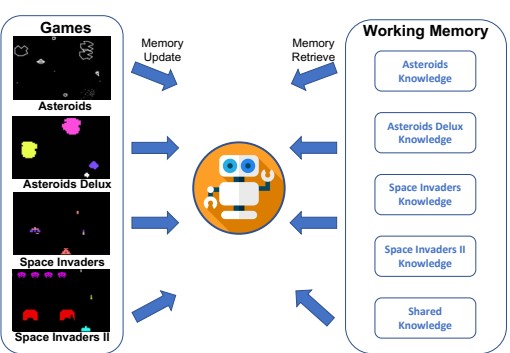

Figure 1: A robot uses its memory to guide its playing strategy.

duces accessibility, and more efficient uses of scale might improve performance further. To address some limits of the implicit, parameter-based memory of large models, we take the inspiration from the concept of "working memory" (Baddeley, 2003; Cowan, 2008) to explicitly store and recall past experiences for use in future decision-making. The concept, "working memory", originates from cognitive psychology and neuroscience (Baddeley, 2003; Goldman-Rakic, 1995), where it refers to the system responsible for the temporary storage and manipulation of information during cognitive tasks.

Our motivation comes from how humans think before they act: they can reason on past experiences to generate appropriate behavior in new situations. As an illustration, imagine we want to train a robot to play four different Atari games: Asteroids, Asteroids Deluxe, Space Invaders, and Space Invaders II (Figure 1). Asteroids Deluxe is a sequel to Asteroids that introduces new boss fights and enemies, and the same can be said about Space Invaders II and Space Invaders. For the robot to play these four games, it must actively store what it has learned in each game in its memory module and choose the

appropriate strategy for each game. Throughout training, the robot's memory module continuously processes and updates relevant game information, allowing it to make informed decisions and adapt its strategies.

Followed by this intuition, we introduce **D**ecision **T**ransformers with **M**emory (DT-Mem): it stores an internal memory as a matrix and its functioning entails two primary steps: **memory update** and **memory retrieval**. DT-Mem builds on earlier work on memory-augmented neural networks (Santoro et al., 2016)—including neural Turing machines (Graves et al., 2014) and memory networks (Sukhbaatar et al., 2015)—in several ways, as we detail in the related work.

We use content-based addressing (Eslami et al., 2016) to locate the memory position to update or retrieve from. The memory update involves modifying or replacing existing information. This enables the system to keep track of changes, maintain task-relevant information, and facilitate decision-making. More specifically, we first map the input sequence and memory into three entities: query, key, and value. Next, we use an attention-based mechanism to calculate the correlations between the input and memory, and then we use the attended weight of the input sequence to update the memory. Memory retrieval refers to the process of accessing and recovering stored information. It involves bringing relevant information back to condition decision-making. To do so, we read from the updated memory at the content-based address.

Since experience must often be mapped from one task to another (e.g., through analogy in humans) to be useful, we also equip our memory module with an adaptable mapping capability. Specifically, for adapting the memory module to a new task, we employ the Low-Rank Adaptation (LoRA) method as described in (Hu et al., 2022) to fine-tune it. The main idea behind LoRA is to train a low-rank projection matrix on a small amount of labeled data from a new task. This matrix maps the parameters of a pre-trained model to a new task. We fine-tune only the memory module in this work because we rely on the generalization capacity of a pre-trained Decision Transformer (DT). Transformers are often pre-trained on large-scale datasets, as in the case of models like Multi-game DT (Lee et al., 2022) and Hyper-DT (Xu et al., 2023), and this pre-training enables them to capture broad knowledge that is transferable across tasks. In contrast, our memory module stores task-specific knowledge that should be adapted for new tasks.

The functioning of DT-Mem differs from external memory and information retrieval-based methods in several ways: (1) memory size, (2) representation of stored information, and (3) retrieval method. In contrast to internal memory module, external memory methods generally require a large dataset that serves as a look-up table. Each raw data point in the external memory also requires an extra step of representation learning to be input to the neural network. Finally, our memory module relies on an attention-based retrieval method, since attention has demonstrated the ability to generalize across tasks. However, attention is computationally impractical for large sets, and hence external/retrieval-based memory systems tend to rely on $k$-nearest neighbor search for information retrieval.

To validate our approach, we evaluate DT-Mem in two environments: (a) on Atari games against Multi-game Decision Transformer (MDT, Lee et al., 2022) and Recurrent Memory Decision Transformer (RMDT, Bessonov et al., 2023), and (b) on Meta-World environments against Prompt Decision Transformer (PDT, Xu et al., 2022) and Hyper-Decision Transformer (HDT, Xu et al., 2023). Our results show that DT-Mem improves generalization and adaptability with fewer model parameters and less training time.

## 2 RELATED WORK

**Transformer-based Reinforcement Learning methods** Transformer (Vaswani et al., 2017) is a powerful architecture designed for sequence modeling. Owing to the capabilities that emerge as model and data size scale up, the Transformer has become a foundational model in several domains, including natural language processing (Brown et al., 2020; OpenAI, 2023; Touvron et al., 2023) and computer vision (Dosovitskiy et al., 2021). However, applying Transformer in reinforcement learning settings, such that it generalizes to multiple tasks, remains an open problem.

Recently, Chen et al. (2021) and Janner et al. (2021) treat the RL problem as a sequence modeling problem and proposed a Transformer-based architecture to solve it with offline RL. These findings inspired researchers to develop more advanced Transformer-based RL methods. Subsequent efforts mainly focus on two aspects: generalization and adaptability. To improve model online adaptabil-

ity, Zheng et al. (2022) propose the Online Decision Transformer (Online DT), which utilizes the maximum-entropy idea to encourage pre-trained policies to explore during a phase of online adaptation. To improve offline adaptation, Xu et al. (2023) propose a Hyper-network-based module that helps DT adapt to unseen tasks efficiently. To facilitate task adaptation, Xu et al. (2022) introduce the prompt-based DT, which selects short trajectories to use in a task prompt in analogy with in-context learning for large language models. Furthermore, Lee et al. (2022) propose a multi-game DT (MDT), which use the expert action inference to consistently produce actions of highly-rewarding behavior. MDT demonstrates that DT can generalize to various Atari games with human-level performance.

We argue that the generalization of the above-mentioned works relies on the size of models and does not learn the data efficiently. To address this issue, we introduce a memory module that can store, blend, and retrieve training information for better model and training efficiency.

**Working memory** In the context of machine learning, there is a long history of neural network-based models that incorporate memory mechanisms (Das et al., 1992; Schmidhuber, 1992; Hochreiter and Schmidhuber, 1997; Santoro et al., 2016; Ba et al., 2016; Munkhdalai and Yu, 2017; Csordás and Schmidhuber, 2019; Ramsauer et al., 2020; Wu et al., 2022a). Generally, this research aims to enhance the capacity of neural networks to store and manipulate information over extended periods of time, leading to improved performance on a range of tasks. It often takes inspiration from human cognitive function. Most salient to our work, Graves et al. (2014) merge concepts from Turing machines and deep learning in "Neural Turing Machines" (NTMs), neural networks that include a content-addressable matrix memory space for storing and updating information throughout time. They show NTMs to be effective for various algorithmic tasks. Concurrently, Sukhbaatar et al. (2015) introduce "memory networks," which use a content-addressable matrix memory store and retrieve information from previous computational steps to facilitate complex reasoning and inference tasks. infinity-former excels in handling unbounded contexts with precision and flexibility, ideal for extensive and complex datasets (Martins et al., 2021). LONGMEM decoupled architecture and token-to-chunk retrieval make it adept at managing large contexts and overcoming memory staleness Wang et al. (2023). kNN-augmented Transformer offers flexibility in context length and rapid adaptation to new data, enhancing the model's real-time applicability Wu et al. (2022b). More recently, Bessonov et al. (2023) introduces a recurrent memory mechanism to address reinforcement learning challenges, which preserves a hidden state throughout the decision-making process. However, this method overlooks the storage and retrieval of task-related information, thereby falling short in fostering model generalization and task adaptation. Munkhdalai et al. (2019) propose a rapidly adaptable neural memory system, which they instantiate as a feedforward neural network trained by metalearning. They evaluate the memory's effectiveness in a simple RL setting, maze exploration, and on various NLP tasks. Alternatively, Goyal et al. (2022) builds on the "global workspace" theory from cognitive science, which posits that different input entities share information through a common communication channel. The proposed shared global workspace method employs the attention mechanism to encourage the most useful information to be shared among neural modules. It is closely related to working memory and inspires us to explore how an explicit working memory can improve the generalization of Transformer-based models. An upshot of our work is that it may be valuable to revisit earlier memory-augmentation methods in light of more powerful foundation models.

# 3 PRELIMINARIES

## 3.1 OFFLINE REINFORCEMENT LEARNING

A trajectory consists of a series of states, actions, and rewards, expressed as $\tau = (s_0, a_0, r_0, s_1, a_1, r_1, \cdots, s_T, a_T, r_T)$. In the context of offline RL, data acquisition doesn't come from active interaction with the environment. Instead, we rely solely on a predefined and limited dataset containing various trajectories generated by different policies. This scenario presents greater challenges as it restricts the agent's ability to actively explore the environment and gather new information, which is a crucial aspect of traditional RL approaches.

Formally, in the context of model evaluation, we can define a set of training tasks and testing tasks as $T^{train}$ and $T^{test}$, respectively. These two sets deliberately have no overlapping tasks, but they may share the same or similar observation and action spaces. To be more specific, for each training task $\mathcal{T}^i \in T^{train}$, we have access to a large training dataset, which contains trajectories

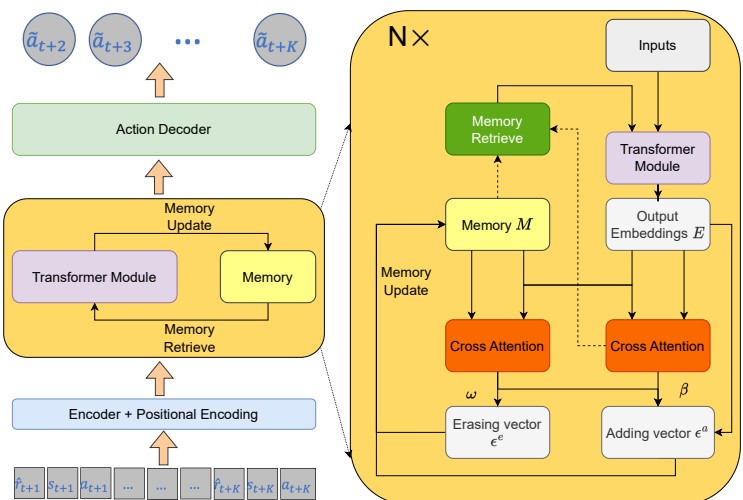

Figure 2: An overview of the proposed DT-Mem architecture.

$\tau^{0:H} = (s_0, a_0, r_0, \cdots, s_H, a_H, r_H)$, where $H$ is the episode length. However, we assume access to only a small amount of data for the testing tasks.

Our goal is to evaluate the proposed model in two dimensions. First, we want to assess the model's **generalization**, which refers to its ability to solve the testing tasks within a finite time with no additional fine-tuning. Second, we want to test the model's **adaptability**, which refers to its ability to improve its performance on the testing tasks through fine-tuning on limited data after pre-training on separate tasks.

### 3.2 LOW-RANK ADAPTATION

Low-rank adaptation (LoRA, Hu et al., 2022) is a transfer learning technique used to adapt a pre-trained model to a new task with limited labeled data. LoRA assumes that the pre-trained model's parameters can be expressed as a low-rank matrix, and that only a small number of parameters must be modified to adapt the model to the new task. The main idea behind LoRA is to utilize a small amount of labeled data from a new task to learn a low-rank projection matrix. This matrix maps the parameters of a pre-trained model to the new task.

## 4 METHODOLOGY

### 4.1 OVERVIEW OF DT-MEM

In Figure 2, we depict the architecture of DT-Mem, which consists of three components: the Transformer module, the Memory module, and the Multi-layer perceptron (MLP) module. The primary role of the Transformer module is to capture dependencies and relationships between states, actions, and returns in a sequence. The input of the Transformer module is a fixed-length sequence of trajectories, denoted as $\tau^{t+1:t+K}$. The output is a sequence of embeddings, where each entry can be attended state embeddings, action embeddings, or return-to-go embeddings. The Transformer module follows the architecture of GPT-2 (Radford et al., 2019), but without the feed-forward layer after attention blocks. We separate the GPT-2 architecture into two pieces: the Transformer module and the MLP module, following the setup for natural language processing tasks: one GPT-2 model can be applied to a wide variety of tasks with different MLP modules Radford et al. (2019). Finally, we introduce a memory module for storing and manipulating intermediate information. This is inspired by the Neural Turing Machine (Graves et al., 2014), where the memory is utilized to infer multiple algorithms.

## 4.2 MEMORY MODULE

The design for the memory module is inspired by the way humans think before they act. Its functioning consists of three parts: identifying salient information output from the transformer module, determining where to store new information and how to integrate it with existing memories, and considering how to use these memories for future decision-making. We have broken down these questions and designed the following steps to address them.

***Step 0: Memory Module Initialization.*** The is initialized as a random matrix $M$, where each row $m_i \in \mathbb{R}^d$, with $i \in [0, N]$, represents a memory slot.

***Step 1: Input Sequence Organizing.*** Initially, we restructure the input sequence to adopt a different format. As illustrated in the problem formulation, the input sequence comprises multiple steps of the tuple $< \hat{r}_t, s_t, a_t >$. Instead of directly feeding this sequence into the transformer module, we treat each tuple as an entity and embed them within the same space. Specifically, we define embedding functions $g_s(s) = e_s$, $g_a(a) = e_a$, and $g_r(\hat{r}) = e_{\hat{r}}$, where $e_s$, $e_a$, and $e_{\hat{r}} \in \mathbb{R}^d$ with $d$ representing the dimension in the latent space. The final input sequence emerges from the concatenation of embeddings $E = [\cdots; e_{s_t}, e_{a_t}, e_{\hat{r}_t}; \cdots]$.

Given our memory structure as a matrix with fixed dimensions (i.e., number of slots ∗ dimensions), it's crucial to synchronize the input dimensions for efficient storage. It's noteworthy that in this design, we maintain the relationships among them as posited in the DT paper, although this is not a requisite. For instance, in the trajectory transformer Janner et al. (2021), states, rewards, and others are grouped individually. As demonstrated in Appendix B.6, these varied designs exhibit no significant difference.

***Step 2: Content-based Address.*** We use an attention-based method to locate the correct memory slot for new input by identifying correlated information. This approach is based on the idea that humans tend to store and group similar information together. To locate the memory position, we utilize an attention mechanism. The position address $w$ is calculated as: $w = \text{softmax}\left(\frac{QK^T}{\sqrt{d}}\right)$. Here, $Q = MW^q$ and $K = EW^k$, where $W^q$ and $W^k$ are parameters for the Multi-layer perceptron (MLP). The objective is to map the memory and input information into the query and key matrix, and then use the dot product to determine the similarities between these two matrices. The softmax function guarantees that the sum of all addresses equals one.

***Step 3: Memory update.*** To store incoming information and blend it with existing memory, we calculate two vectors: an erasing vector, $\epsilon^e$, and an adding vector, $\epsilon^a$. The erasing vector erases the current memory, while the adding vector controls information flow to the memory. To achieve this goal, we again utilize the attention mechanism. First, we map memory and input information to query, key, and value vectors, denoted as $\hat{Q} = M\hat{W}^q$, $\hat{K} = E\hat{W}^k$, and $\hat{V} = E\hat{W}^v$, respectively, where $\hat{W}^q$, $\hat{W}^k$, and $\hat{W}^v$ are parameters. Next, we calculate the writing strength, $\beta = \text{softmax}\left(\frac{\hat{Q}\hat{K}^T}{\sqrt{d}}\right)$. The erasing vector is used to selectively erase information from the memory matrix and is computed as a function of the content-based addressing vector and the write strength. The erasing vector is calculated as $\epsilon^e = w \odot (1 - \beta)$, where $\odot$ indicates element-wise multiplication. The complement of the write strength is 1 minus the write strength, so this will result in a vector where the elements corresponding to the selected memory locations are set to 0, and the elements corresponding to the unselected memory locations are unchanged.

The adding vector is used to selectively add information to the memory matrix and is computed as a function of the write strength and the input vector. Specifically, the adding vector is calculated as $\epsilon^a = (w \odot \beta)\hat{W}^v x$.

Finally, the memory is updated as $M_t = M_{t-1} \odot (1 - \epsilon^e) + \epsilon^a$. If the selected memory slot is empty or erased, the new information will be stored. Otherwise, the new information will be blended with the existing memory contents.

***Step 4: Memory retrieve*** To utilize memory for decision-making, we retrieve information from the updated memory slot. Reading from the memory matrix is done by computing a read position vector. This vector can be computed using the above content-based addressing mechanism that compares the query vector with the contents of the memory matrix. Note that in other retrieval-based methods (Humphreys et al., 2022; Borgeaud et al., 2022), the nearest neighbor is the common way to retrieve related information. However, in our case, the internal memory is smaller than the typical external

memory, which makes attention-based retrieval feasible. Since the query information is the same as the input information, we use the same content address to retrieve the memory: $\boldsymbol{E}_{out} = \boldsymbol{w} \odot \boldsymbol{M}_t$.

### 4.3 PRE-TRAINING DT-MEM

We use a set of training tasks $T^{train}$, where each task $\mathcal{T}_i \in T^{train}$ has an associated offline dataset $\mathcal{D}_i$ consisting of hundreds of trajectories $\tau$ generated by a behavior policy. The behavior policy can be either a pre-trained policy (such as DQN) or a rule-based policy, depending on what is available. Each trajectory $\tau = (s_0, a_0, r_0, \cdots, s_H, a_H, r_H)$, where $s_i \in \mathcal{S}, a_i \in \mathcal{A}, r_i \in \mathcal{R}$, and $H$ is the episode length.

To serve as an input to the DT-Mem, we first segment the trajectory $\tau$ into several pieces, each with length $K$. We denote $\tau_{t+1:t+K} = (s_{t+1}, a_{t+1}, r_{t+1}, \cdots, s_{t+K}, a_{t+K}, r_{t+K})$ as one of the input sequence. However, we modify these trajectories instead of inputting them directly. Specifically, we follow the return-to-go Decision Transformer idea Chen et al. (2021) and calculate the return to go, $\hat{r}_t = \sum_{t+1}^{t+K} r_t$, for every timestep. This is effective because $\hat{r}_t$ acts as a subgoal. It encourages the Transformer module to generate actions that can reduce the negative of this value as close to zero as possible. Then we input the modified trajectories $\hat{\tau}_{t+1:t+K} = (\hat{r}_{t+1}, s_{t+1}, a_{t+1}, \cdots, \hat{r}_{t+K}, s_{t+K}, a_{t+K})$ to the transformer module. The output of the transformer module is a sequence embedding $e_{seq} \in \mathbb{R}^{d \times 3K}$, where $d$ is the dimension of the embedding space.

Next, we transmit $e_{seq}$ to the Memory module to update and retrieve the memory information. Finally, we use the retrieved memory $\boldsymbol{E}_{out}$ and MLP modules to generate the corresponding actions $\hat{a}_t$. We minimize a supervised training loss with three terms: predicted actions $\tilde{a}_t$, predicted reward $\tilde{r}_t$, and predicted return-to-go $\tilde{R}_t$. The loss function is:

$$\mathcal{L} = \sum_{t+1}^{t+K} ||\tilde{a}_t - a_t||^2 + \alpha ||\tilde{r}_t - \hat{r}_t||^2 + \lambda ||\tilde{R}_t - r_t||^2, \tag{1}$$

where $\alpha$ and $\lambda$ are scalar hyper-parameters. In experiments, we find that the final performance is not sensitive to these two hyper-parameters, so we set them to 1 for simplicity.

The full pre-training process is summarized in Appendix A.3 Algorithm 1.

### 4.4 FINE-TUNING DT-MEM WITH LoRA

Fine-tuning LLMs involves heavy computation due to the large number of parameter updates required. We argue that fine-tuning only the memory module can achieve results comparable to those of fine-tuning the entire parameter space. LLMs benefit from being trained on large-scale datasets, which expose the model to a diverse range of linguistic patterns and semantic relationships, such as models like (Devlin et al., 2019) or GPT (Radford et al., 2019). This exposure helps the model learn robust and generalized representations that can capture different aspects of language understanding and generation. After pre-training, the model can be fine-tuned on specific downstream tasks with task-specific labeled data. In our case, this task-specific knowledge is stored in the memory module. Thus, fine-tuning the memory module helps the model update its memory module to adapt to the new task.

We apply the low-rank adaptation approach (LoRA, Hu et al., 2022) to fine-tune the memory module. Specifically, we modify the forward pass by adding low-rank matrices to $\boldsymbol{W}^q$, $\boldsymbol{W}^k$, $\boldsymbol{W}^v$, $\hat{\boldsymbol{W}}^q$, and $\hat{\boldsymbol{W}}^k$. Let's take $\boldsymbol{W}^q$ as an example. Assuming the original output for query information is $\boldsymbol{Q} = \boldsymbol{M}\boldsymbol{W}^q$, we adapt this query value to a new task as $\boldsymbol{Q}' = \boldsymbol{M}(\boldsymbol{W}^q + \boldsymbol{B}^q\boldsymbol{A}^q)$, where $\boldsymbol{W}^q \in \mathbb{R}^{n \times d}, \boldsymbol{B} \in \mathbb{R}^{n \times m}$, and $\boldsymbol{A} \in \mathbb{R}^{m \times d}$, and $m$ is the size of the memory module. Since the rank $m \ll min(n, d)$, fine-tuning the parameters $\boldsymbol{B}^q$ and $\boldsymbol{A}^q$ reduces the number of trainable parameters for downstream tasks. We perform supervised training by computing the loss between the model's output and the labels in the fine-tuning dataset. During this process, only $\boldsymbol{B}^q$ and $\boldsymbol{A}^q$ are updated. The detailed fine-tuning procedure can be seen in Appendix A.3 Algorithm 2.

## 5 EVALUATION

We design our experiments to answer the following questions: **Q1:** Does DT-Mem improve model generalization? **Q2:** Does DT-Mem improve pre-training results and training efficiency? **Q3:** Does DT-Mem scales with model size? **Q4:** Does fine-tuning only the memory module improve model adaptability?

Recall that we use generalization to refer to performance on tasks the model has never trained on (zero-shot), and adaptability to refer to performance after fine-tuning.

### 5.1 ENVIRONMENTS AND MODELS SETUP

**Atari Games** To ensure a fair comparison with the Multi-Game Decision Transformer, we used the same Atari dataset, which comprises multiple training runs of DQN trajectories. Due to limited compute resources and to prevent cherry-picking, we select 17 games from the available 41 based on their alphabetical order, as introduced in Lee et al. (2022). For each game, the data contains 50 policy checkpoints, each containing 500k environment steps. For the fine-tuning dataset, we randomly selected 10% of the data from the unseen dataset, which yielded 50k environment steps. Following the settings from Lee et al. (2022), we choose five games (Alien, Ms. Pac-Man, Pong, Space Invaders, and Star Gunner) to be used only for fine-tuning. Moreover, Brandfonbrener et al. (2022) suggests that return-conditioned supervised learning (RCSL) algorithms require strong dataset coverage to select a near-optimal policy. Therefore, our dataset contains both expert and non-expert behaviors.

**Meta-World** To make a fair comparison with Hyper-DT and Prompt-DT, we evaluate the proposed method on the Meta-World environment (Yu et al., 2019). We evaluate using the Meta-World ML45 benchmark, which includes 45 training tasks and 5 testing tasks. Following the approach taken in Xu et al. (2023), for each training task, we generat an offline dataset containing 1000 episodes for each game, using a rule-based script policy. For fine-tuning data, we randomly pick 10k episodes from the testing dataset, as compared to 20k-80k episodes used in Hyper-DT.

**DT-Mem settings** We report results for DT-Mem 20M (20 million parameters), which consists of 13M transformer parameters and 7M memory module parameters. We specify the architecture completely in Appendix A.1.

**Training and Fine-tuning** For all games, we use eight V100 GPUs for model training and one V100 GPU for fine-tuning. We train on both Atari games and Meta-World for 10M steps. For fine-tuning on unseen scenarios, we train for 100k steps.

### 5.2 BASELINE METHODS

We compare DT-Mem's performance against the following baselines. **MDT** Multi-game Decision Transformer (Lee et al., 2022), which trains a large transformer-based model on multi-game domains. For a fair comparison, we train an MDT with 20M parameters, which is approximately the same size of DT-Mem. **RMDT** Recurrent Memory Decision Transformer (Bessonov et al., 2023), which utilizes a recurrent memory mechanism for solving reinforcement learning problems. This is the most related memory-based DT that is close to our work. **HDT** Hyper-Decision Transformer (Xu et al., 2023), which utilizes a hyper-network module to help DT adapt rapidly to unseen tasks. Since we do not have access to the implementation at the time of writing, for the sake of correctness, we compare our model with HDT on Meta-World only. The results reported in our evaluation section come from the HDT paper. **PDT** The Prompt Decision Transformer (Xu et al., 2022) generates actions by considering both recent context and pre-collected demonstrations from the target task.

### 5.3 DT-MEM IMPROVES MODEL GENERALIZATION.

We evaluate five held-out games fine-tuning results as listed in Table 1. Each evaluation signifies an average derived from 16 runs, each under differing random seeds. The derived results show that the memory-incorporated method, RMDT and DT-Mem, enhances model generalization when compared to their ablation method MDT. A noteworthy observation is that DT-Memdemonstrates superior generalization performance than RMDT in four out of the five games. Neither of the methods achieves a good result in "Pong". We further discuss whether fine-tuning helps to improve the performance in Section 5.5.

| | Alien | MsPacman | Pong | SpaceInvaders | StarGunner |
|---|---|---|---|---|---|
| MDT | 3.8% (±0.4%) | 13.2% (±1.3%) | 0% (±0%) | 8.6% (±1.6%) | 2.3% (±0.1%) |
| RMDT | 22.3% (±10.7%) | 22.9% (±8.9%) | 0% (±0%) | 17.6% (±9.2%) | 27.7% (±11.5%) |
| DT-Mem | 51.0% (±32.2%) | 69.3% (±19.3%) | 0% (±0%) | 53.6% (±29.0%) | 62.2% (±19.1%) |

Table 1: Evaluation results on 5 held-out games after pre-training on other Atari Games. Each value represents the DQN-normalized score, computed with a 95% confidence interval.

## 5.4 DT-MEM ENABLES MORE COMPUTATIONALLY EFFICIENT TRAINING AND SCALE WITH MODEL PARAMETERS.

To demonstrate training efficiency, we illustrate the model training time in Table 4 and the training curve in Appendix B.2 Figure 7. During training, we find that DT-Mem reduces the training time by approximately 4 times, 8 times, and 32 times compared to MDT-13M, MDT-40M, and MDT-200M, respectively. For the training curve, it is reasonable to report the prediction loss on the training dataset since we use a supervised loss. Here, the prediction accuracy consists of three parts: action prediction accuracy, reward prediction accuracy, and return prediction accuracy.

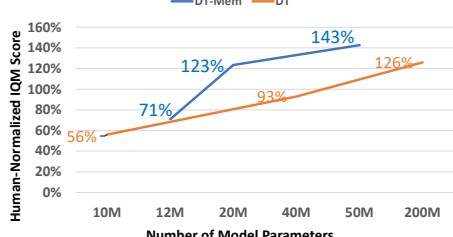

Figure 3: Scaling of IQM scores

| Model | Training time (hours) |
|---|---|
| **DT-Mem** | **50** |
| MDT-13M | 200 |
| MDT-40M | 400 |
| MDT-200M | 1600 |

Figure 4: Model training time

Figure 3 showcases the scaling laws of the proposed DT-Mem model. We measure performance using the human-normalized IQM score. It's crucial to note that for all instances of DT-Mem, we maintained a consistent number of memory slots. From the result, it's evident that the performance of DT-Mem scales with the number of parameters. Notably, the generalization of DT-Mem with 20M parameters is approximately on par with the 200M parameter version of MDT. Furthermore, the 50M DT-Mem surpasses MDT by a margin of 16.7%.

## 5.5 FINE-TUNING ONLY THE MEMORY MODULE IMPROVES MODEL ADAPTABILITY.

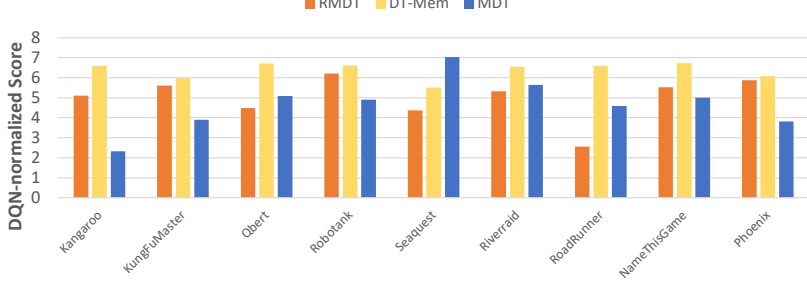

Figure 5: Fine-tuning performance on 10% of dataset in unseen Atari games. For better visualization, the y-axis is the logarithm of DQN-normalized score.

Another question we care about is how the pre-trained DT-Mem performs on unseen tasks. We randomly selected nine unseen Atari games and evaluated their performance through relative im-

provement scores, as shown in Figure 5. DT-Mem consistently outperforms RMDT and MDT in most of the games listed, with the exception of Seaquest, where MDT excels. MDT exhibits the least superior performance across most games, with its performance particularly lagging in KungFuMaster, Robotank, and Phoenix. RMDT holds an intermediate performance level between DT-Mem and MDT across most games. The consistent superior performance of DT-Mem across most games suggests that this method might have a more adaptable approach. The singular superior performance of MDT in Seaquest prompts a further investigation into the unique attributes of this game that may favor the MDT method.

To further understand the adaptability of the proposed method, we compare DT-Mem with HDT and PDT in meta-world environments. The quantitative fine-tuning results are shown in Table 2. Overall, DT-Mem achieves the best performance in the comparison. As we can see, compared to HDT, DT-Mem increases training, testing (no-FT), and testing (FT) scores by an average of 3%, 8%, and 3%, respectively. Moreover, the HDT adaptation module (hyper-network module), while small (69K) relative to the full model (13M), relies on the pre-trained hyper-network, which contains 2.3M parameters. We argue that the hyper-net is more burdensome than our design: it uses more than 10x the number of adaptation parameters (147K) used by DT-Mem and requires an extra compute phase to pre-train the hyper-network module.

| | Model Sizes | | Meta-World ML45 Performances | | |
|---|---|---|---|---|---|
| | Adaptation | Percentage | Train | Test (no-FT) | Test (FT) |
| HDT | 69K | 0.5% | $0.89 \pm 0.00$ | $0.12 \pm 0.01$ | $0.92 \pm 0.10$ |
| PDT | 6K | 0.05% | $0.88 \pm 0.00$ | $0.06 \pm 0.05$ | $0.09 \pm 0.01$ |
| DT-Mem | 147K | 0.7% | $\mathbf{0.92 \pm 0.00}$ | $\mathbf{0.20 \pm 0.01}$ | $\mathbf{0.95 \pm 0.10}$ |

Table 2: Evaluation results on Meta-World ML45 benchmarks

### 5.6 DT-MEM IMPROVES TRAINING PERFORMANCE.

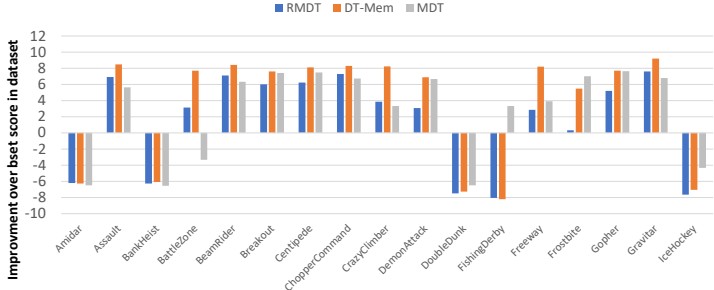

Figure 6: The percent improvement for training dataset.

In this section, we evaluate whether adding the memory module helps improve the pre-training performance. Thus, we choose relative improvement: rel-imp(%) = (model score − best score in data/best score in data × 100 to measure the model performance. For better visualization, we take the logarithm of the rel-imp(%). As shown in Figure 6, the proposed DT-Memout performs MDT in 13 out of 17 games. DT-Mem outperforms RMDT in 15 out of 17 games. These results demonstrates that memory module improves the policy training performance.

## 6 CONCLUSION

LLM-based RL algorithms have shown generalization across multiple tasks and games. We argue that this ability comes from implicit memory that fits a large number of parameters to the training data, which is inefficient in terms of model size. In contrast, we propose a new approach inspired by the concept of "working memory" called **D**ecision **T**ransformers with **M**emory (DT-Mem), which stores training experience explicitly in a content-addressable matrix module for later retrieval and use. The evaluation demonstrates that DT-Mem achieves better generalization on Atari games with only 10% of the model parameters compared to the state-of-the-art method. We also show that DT-Mem outperform other memory-based DT methods in terms of generalization and adaptability. Furthermore, we demonstrate that fine-tuning DT-Mem with a small amount of data can produce state-of-the-art results on both Atari games and the Meta-World environment, when compared to MDT, RMDT, PDT, and HDT.

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
