## A  IMPLEMENTATION DETAILS

### A.1  DT-MEM NETWORK ARCHITECTURE

Table 3 summarizes the different model configurations used for evaluation. In this section, we describe these model configurations in detail. While Table 3 provides a summary, we will also provide additional information here. DT-Mem, PDT and HDT are all share the same transformer architectures. However, for task-adaptation, HDT utilizes a pre-trained 2.3M hyper-network, while DT-Mem introduces 147K LoRA parameters. To compare with MDT, we use the same parameter size as reported in Lee et al. (2022).

| Model | Layers | Hidden size (d) | Heads | Params | Memory Size | Memory Module Params |
|-------|--------|-----------------|-------|--------|-------------|----------------------|
| HDT | 4 | 512 | 8 | 13M | N.A. | N.A. |
| MDT-200M | 10 | 1280 | 20 | 200M | N.A. | N.A. |
| DT-Mem | 4 | 512 | 8 | 13M | 559K | 7M |

Table 3: Detailed Model Sizes

### A.2  HYPER-PARAMETERS

In this section, we will delve into the specifics of the model parameters. Understanding these parameters is key to understanding the workings of the model. It is worth noting that the source code for this model is publicly available at https://anonymous.4open.science/r/DT-Mem-Submission277/README.md. This allows for a deeper understanding of the model's inner workings and may facilitate the replication of its results.

| Hyperparameters | Value |
|-----------------|-------|
| K (length of context) | 28 |
| dropout rate | 0.1 |
| maximum epochs | 1000 |
| steps for each epoch | 1000 |
| optimizer learning rate | 1e-4 |
| weight decay | 1e-4 |
| gradient norm clip | 1. |
| data points for each dataset | 500,000 |
| batch size | 64 |
| memory slots | 1290 |
| activation | GELU |
| optimizer | AdamW |
| scheduler | LambdaLR |

Table 4: Hyperparameters for DT-Mem training

### A.3  TRAINING AND FINE-TUNING ALGORITHM

In this section, we present the pre-training DT-Memin Appendix A.3 and fine-tuning DT-Mem with LoRA in Appendix 5.5.

We pre-train DT-Mem on multiple offline datasets. Each gradient update of the DT-Memmodel considers information from each training task.

We fine-tune the memory module to adapt to each downstream task. To achieve this, we fix the pre-trained DT-Mem model parameters and add additional LoRA parameters for the memory module feed-forward neural networks. The fine-tuning dataset is used to update these LoRA parameters only.

---

**Algorithm 1** Pre-train DT-Mem

---

1: **for** T episodes **do**
2:     **for** Task $\mathcal{T}_i \in T^{train}$ **do**
3:         Sample trajectories $\tau = (s_0, a_0, r_0, \cdots, s_H, a_H, r_H)$ from the dataset $\mathcal{D}_i$.
4:         Split trajectories into different segments with length K and calculate return-to-go in the input sequence.
5:         Given $\hat{\tau}_{t+1:t+K}$, compute the sequence embedding $e_{seq}$.
6:         Update the memory module and retrieve the relative information as $\boldsymbol{E}_{out}$
7:         Given $\boldsymbol{E}_{out}$, predict actions $\tilde{a}_t$, reward $\tilde{r}_t$, and return-to-go $\tilde{R}_t$.
8:         Compute the loss according to Eqn. 1.
9:         Update all module parameters.
10:     **end for**
11: **end for**

---

**Algorithm 2** Fine-tuning DT-Mem

---

**Require:** Fine-tuning dataset $\mathcal{T}^i \in T^{test}$ dataset $\mathcal{D}^i$ for $\mathcal{T}^i$. Initialize LoRA parameters $\hat{\boldsymbol{B}}^q, \hat{\boldsymbol{B}}^k, \hat{\boldsymbol{B}}^v, \hat{\boldsymbol{A}}^q, \hat{\boldsymbol{A}}^k, \hat{\boldsymbol{A}}^v, \boldsymbol{B}^q, \boldsymbol{A}^q, \boldsymbol{B}^k, \boldsymbol{A}^k$.

1: **for** T steps **do**
2:     Split trajectories into different segments with length K and calculate return-to-go in the input sequence.
3:     Given $\hat{\tau}_{t+1:t+K}$, compute the sequence embedding $e_{seq}$.
4:     Update memory module using $\hat{\boldsymbol{Q}} = \boldsymbol{M}(\hat{\boldsymbol{W}}^q + \hat{\boldsymbol{B}}^q\hat{\boldsymbol{A}}^q)$, $\hat{\boldsymbol{K}} = \boldsymbol{M}(\hat{\boldsymbol{W}}^k + \hat{\boldsymbol{B}}^k\hat{\boldsymbol{A}}^k)$, $\hat{\boldsymbol{V}} = \boldsymbol{M}(\hat{\boldsymbol{W}}^v + \hat{\boldsymbol{B}}^v\hat{\boldsymbol{A}}^v)$, $\boldsymbol{Q} = \boldsymbol{M}(\boldsymbol{W}^q + \boldsymbol{B}^q\boldsymbol{A}^q)$, $\boldsymbol{K} = \boldsymbol{M}(\boldsymbol{W}^k + \boldsymbol{B}^k\boldsymbol{A}^k)$
5:     Retrieve the relative information as $\boldsymbol{E}_{out}$
6:     Given $\boldsymbol{E}_{out}$, predict actions $\tilde{a}_t$, reward $\tilde{r}_t$, and return-to-go $\tilde{R}_t$.
7:     Compute the loss according to Eqn. 1.
8:     Update LoRA parameters only.
9: **end for**

---

---

**Algorithm 3** Memory Operations

---

1: **Step 0: Memory Module Initialization**
2: Initialize memory as a random matrix $\boldsymbol{M}$ where each row $\boldsymbol{m_i} \in \mathbb{R}^d$ and $i \in [0, N]$.
3:
4: **Step 1: Input Sequence Organizing**
5: Restructure input sequence into format $< \hat{r}_t, s_t, a_t >$.
6: Define embedding functions $g_s(s) = e_s$, $g_a(a) = e_a$, $g_r(\hat{r}) = e_{\hat{r}}$.
7: Concatenate embeddings to form input sequence $\boldsymbol{E} = [\cdots ; \boldsymbol{e}_{s_t}, \boldsymbol{e}_{a_t}, \boldsymbol{e}_{\hat{r}_t} ; \cdots]$.
8:
9: **Step 2: Content-based Address**
10: Use attention to locate memory slot for new input.
11: Calculate position address $\boldsymbol{w} = \text{softmax}\left(\frac{\boldsymbol{Q}\boldsymbol{K}^T}{\sqrt{d}}\right)$.
12: Define $\boldsymbol{Q} = \boldsymbol{M}\boldsymbol{W}^q$ and $\boldsymbol{K} = \boldsymbol{E}\boldsymbol{W}^k$.
13:
14: **for** N Times memory operations do **do**
15:     **Step 3: Memory Update**
16:     Calculate erasing vector $\boldsymbol{\epsilon}^e$ and adding vector $\boldsymbol{\epsilon}^a$.
17:     Define $\hat{\boldsymbol{Q}} = \boldsymbol{M}\hat{\boldsymbol{W}}^q$, $\hat{\boldsymbol{K}} = \boldsymbol{E}\hat{\boldsymbol{W}}^k$, $\hat{\boldsymbol{V}} = \boldsymbol{E}\hat{\boldsymbol{W}}^v$.
18:     Compute writing strength $\beta = \text{softmax}\left(\frac{\hat{\boldsymbol{Q}}\hat{\boldsymbol{K}}^T}{\sqrt{d}}\right)$.
19:     Calculate $\boldsymbol{\epsilon}^e = \boldsymbol{w} \odot (1 - \beta)$.
20:     Calculate $\boldsymbol{\epsilon}^a = (\boldsymbol{w} \odot \beta)\hat{\boldsymbol{W}}^v \boldsymbol{x}$.
21:     Update memory $\boldsymbol{M}_n = \boldsymbol{M}_{n-1} \odot (\boldsymbol{1} - \boldsymbol{\epsilon}^e) + \boldsymbol{\epsilon}^a$.
22:
23:     **Step 4: Memory Retrieve**
24:     Retrieve information from memory for decision-making.
25:     Compute read position vector using content-based address.
26:     Retrieve memory $\boldsymbol{E}_{out} = \boldsymbol{w} \odot \boldsymbol{M}_n$.
27:     $\boldsymbol{E} = \boldsymbol{E}_{out}$
28: **end for**
29: output $\boldsymbol{E}$ for action decoder.

---

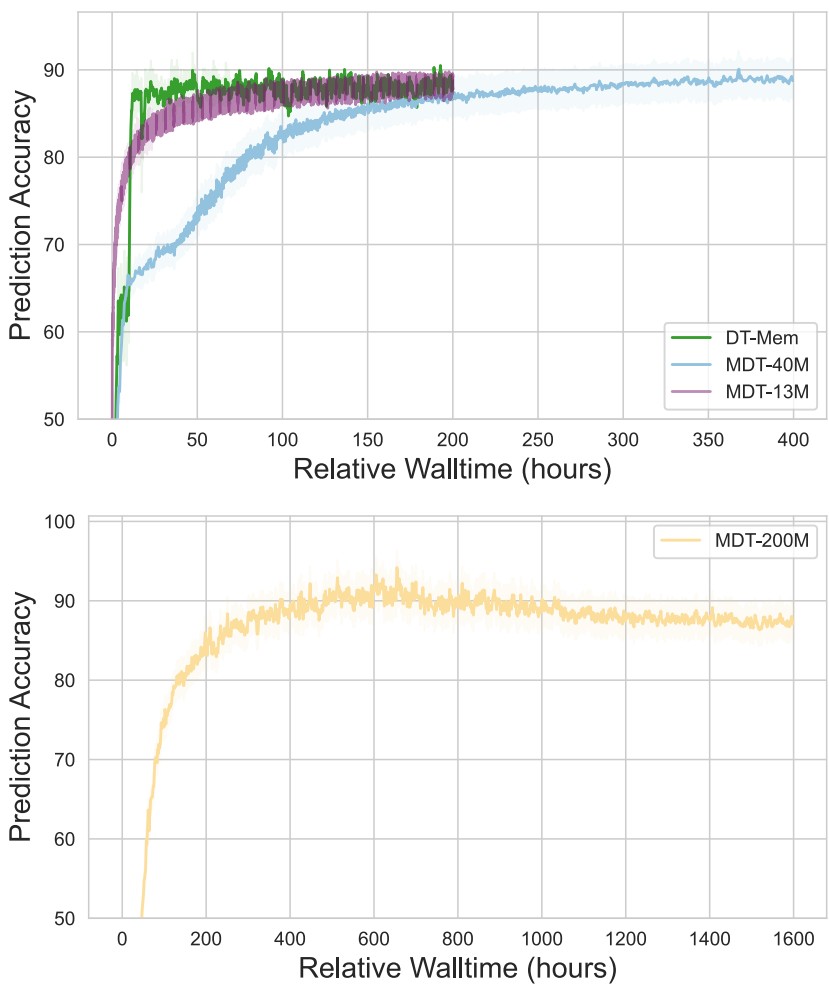

Figure 7: This graph shows the prediction accuracy during training. Each curve represents three runs with different random seeds. For better visualization, MDT-200M is displayed in a separate figure.

## B ADDITIONAL EXPERIMENTS

### B.1 EVALUATION PARAMETERS

To evaluate the performance of our model on Atari games, we randomly selected 16 different random seeds for evaluation. We chose the random seed by multiplying the number of runs by 100. For example, the random seed for run 6 is $6 \times 100 = 600$.

### B.2 TRAINING EFFICIENCIES

To demonstrate training efficiency, we illustrate the model training curve in Figure 7. For the training curve, it is reasonable to report the prediction loss on the training dataset since we use a supervised loss. Here, the prediction accuracy consists of three parts: action prediction accuracy, reward prediction accuracy and return prediction accuracy. The y-axis shows the average value of these three predictions, and the x-axis is the relative walltime based on same computing resources.

### B.3 THE ANALYSIS OF MEMORY SIZE

In this section, we investigate the impact of the memory module size on the performance of DT-Mem. We employ the Bayes optimization strategy to tune the parameters. It's worth noting that the memory

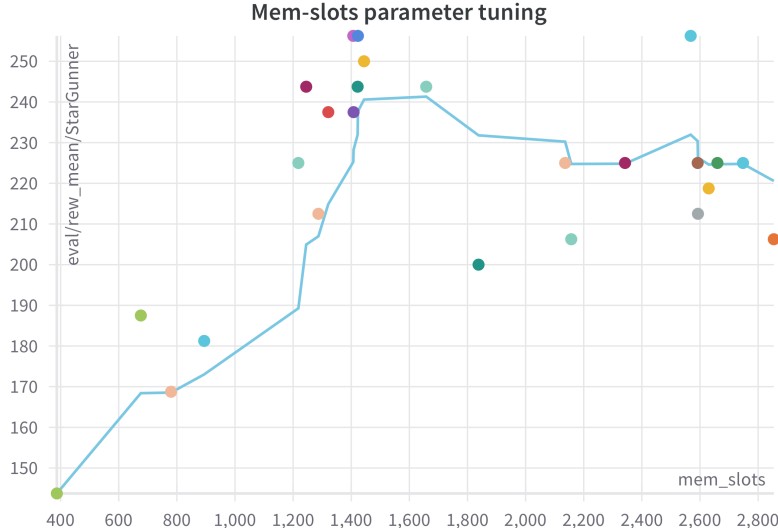

Figure 8: The parameter tuning results for the number of memory slots. The blue curve shows the like from left to right over the x axis and plots the running average y value.

size is calculated by multiplying the number of memory slots by the size of each slot, which is fixed at 512 dimensions for the sake of evaluation simplicity. To expedite the hyper-parameter tuning process, we present the evaluation results based on 100k training steps of the StarGunner game. We assess various configurations of memory slots and calculate their corresponding average rewards over 16 runs. Figure 8 reveals several key findings: (1)Increasing the size of memory slots leads to a higher reward accumulation. Notably, there is a significant performance boost when the number exceeds 1200. (2)In summary, when the number of memory slots exceeds 1800, the performance of the system decreases. This decline occurs because there is a trade-off between the number of memory slots and the training steps. With a larger number of memory slots, it becomes necessary to allocate more training time.

## B.4 ABLATION STUDY OF LORA ADAPTOR

| | Meta-World ML45 Performances | | | Data size | Model | |
| | Train | Test (no-FT) | Test (FT) | | Adap. | Per. |
|---|---|---|---|---|---|---|
| DT-Mem (hyper-net) | $0.92 \pm 0.01$ | $\mathbf{0.23 \pm 0.10}$ | $0.81 \pm 0.15$ | 30 | 5.7M | 43.8% |
| DT-Mem | $\mathbf{0.92 \pm 0.00}$ | $0.20 \pm 0.01$ | $\mathbf{0.95 \pm 0.10}$ | 10 | 147K | 0.7% |

Table 5: Ablation study results on Meta-World ML45 benchmarks. DT-Mem (hyper-net) denotes the variation of DT-Mem, which substitute LoRA adaptation module with hyper-networks. Adap. stands for adaptation parameters, and Per. stands for percentage of original model.

In this section, we conduct an ablation study of LoRA-based memory adaptor. We substitute LoRA adaptor with hyper-networks. Specifically, the parameters of the memory module are generated from hyper-networks. This approach is based on von Oswald et al. (2020), where hyper-networks take task-related information as input and generate the corresponding networks for the downstream MLP. We use the same approach and generate parameters that are conditioned on two types of inputs: the task embedding from the task encoder and the sequence embeddings from the Transformer module.

To generate task embeddings, we adopt the same idea from PDT (Xu et al., 2022), which demonstrates that a small part of trajectories can represent the task-related information. We further extend this idea to fully extract the task information. To achieve this goal, we use a Neural Networks (NNs) as a task encoder. Specifically, this task encoder is implemented as a transformer encoder-like structure Vaswani et al. (2017). We first formulate the first $i$ steps of collected trajectories

$\tau_{0:i} = (s_0, a_0, r_0, \cdots, s_i, a_i, r_i)$ as a task specific information. The task trajectory $\tau_{0:i}$ is treated as a sequence of inputs to the task encoder. The output of the task encoder is a task embedding $e_{task} \in \mathbb{R}^d$, where $d$ is the dimension of the embedding.

Then, we concatenate the task embedding and sequence embedding $e = [e_{task}; e_{seq}]$ and input them to the hyper-networks. Specifically, we define the hyper-network as a function of $f_\omega(\cdot)$ parameterized by $\omega$. The output $\Theta = f_\omega(e)$ is a set of parameters for the memory module.

According to the evaluation results in Table 5, the inclusion of a hyper-network in the DT-Memmodel improves generalization without the need for fine-tuning. However, it is worth noting that the hyper-network variant of DT-Mem(hyper-net) exhibits higher variance compared to DT-Mem. The primary reason for this higher variance is the uncertainty arising from the task information. In each run, different task-related sequences are collected, resulting in varying generated parameters for the memory module. Regarding the task fine-tuning results, we observe that the LoRA module outperforms other methods. This finding indicates that fine-tuning with LoRA enhances the model's adaptability. We hypothesize that the size of the hyper-networks model plays a role in these results. Fine-tuning a large model size (5.7M) with a small step-size (100k steps in our case) becomes challenging. In an effort to improve hyper-networks fine-tuning performance, we increased the fine-tuning dataset from 10k episodes to 30k episodes. These findings suggest that LoRA-based fine-tuning demonstrates better data efficiency.

The motivations for using LoRA to fine-tune the model can be concluded in the following two reasons:

Hu et al. (2022) suggests that the LoRA method, in contrast to other adapters, maintains model quality without introducing inference latency or shortening input sequence length. Furthermore, it facilitates rapid task-switching in service deployments by sharing most model parameters. Parameter-efficient fine-tuning (PEFT) refines a limited number of model parameters, preserving most of the pre-trained LLM parameters, which reduces computational and storage demands (Hu et al., 2022). This approach also addresses catastrophic forgetting [4] and has outperformed standard fine-tuning in low-data and out-of-domain situations [5]. Besides, the results of full parameter fine-tuning vs. PEFT are shown in Table 6:

| Game | PEFT | FFT-Single | FFT-All |
|---|---|---|---|
| Alien | 127.4% | 116.8% | 113.9% |
| MsPacman | 130.8% | 122.8 | 77.1% |
| Pong | 97.8% | 93.7% | 90.5% |
| SpaceInvaders | 100.8% | 86.8% | 73.4% |
| StarGunner | 158.3% | 55.7% | 40.6% |

Table 6: Performance comparison of PEFT across various games

where PEFT stands for LoRA fine-tuning for all games together, FFT-single means full-parameter fine-tuning on a single game only, FFT-All stands for full-tine-tuning on all games together. Results are DQN-normalized score.

## B.5 LoRA HYPER-PARAMETERS TUNING

In this section, we explore the impact of LoRA hyper-parameters on the final fine-tuning results. LoRA employs three hyper-parameters: rank, lora_dropout, and lora_alpha. The rank parameter, denoted as $m$, determines the low-rank of adaptation matrices $\boldsymbol{B} \in \mathbb{R}^{n \times m}$ and $\boldsymbol{A} \in \mathbb{R}^{m \times d}$, as described in Section 4.4. The lora_dropout refers to the dropout rate applied to the LoRA neural networks, while lora_alpha controls the scaling factor of the LoRA outputs. Figure 9 presents the fine-tuning results, with the last column (**eval/rew_mean/StarGur**) specifically showcasing the fine-tuning results for the StarGunner game. To obtain the optimal set of parameters, we employ the Bayesian optimization method for parameter tuning, which suggests various parameter combinations that maximize the fine-tuning results.

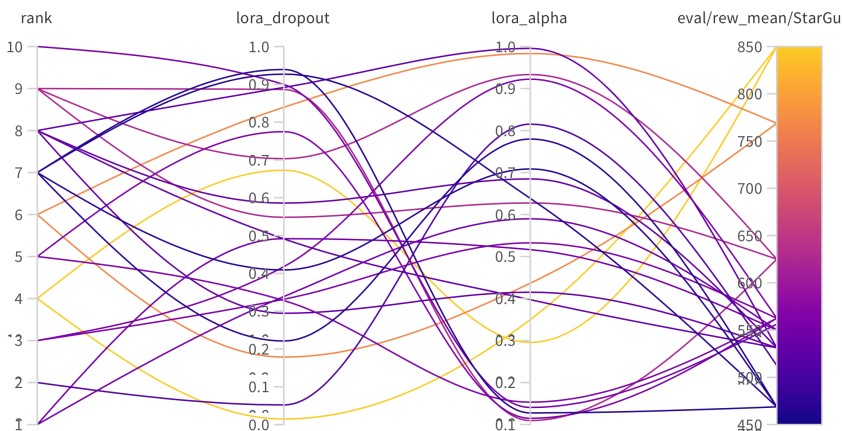

Figure 9: LoRA hyper-parameters tuning results.

| Parameter | Importance score | Correlation score |
|---|---|---|
| rank | **0.486** | -0.132 |
| lora_dropout | 0.285 | -0.561 |
| lora_alpha | 0.229 | 0.550 |

Table 7: Analysis of LoRA hyper-parameters

We further analyze these parameters and present the findings in Table 7. To gain insights, we utilize two widely used metrics in the MLOps platform Weights&Biases[1].

Regarding the **importance score**, we train a random forest model with the hyper-parameters as inputs and the metric as the target output. We report the feature importance values derived from the random forest. This hyper-parameter importance panel disentangles complex interactions among highly correlated hyper-parameters. It facilitates fine-tuning of hyper-parameter searches by highlighting the hyper-parameters that significantly impact the prediction of model performance.

The **correlation score** represents the linear correlation between each hyper-parameter and the chosen metric (in this case, val_loss). A high correlation indicates that when the hyper-parameter has a higher value, the metric also tends to have higher values, and vice versa. Correlation is a useful metric, but it does not capture second-order interactions between inputs and can be challenging to compare when inputs have widely different ranges.

As shown in Table 7, rank emerges as the most important hyper-parameter that requires careful tuning. The correlation score of rank is -0.132, indicating that a smaller rank number leads to better fine-tuning results. Based on our findings, a rank value of 4 yields the best outcome. Lora_dropout and lora_alpha exhibit similar importance scores, suggesting that these two parameters can be treated equally. The correlation score reveals that a smaller lora_dropout value and a larger lora_alpha value result in improved performance.

## B.6 ABLATION STUDIES ON DIFFERENT INPUT SEQUENCE ORGANIZING CHOICES

We examine two distinct approaches to input organization. The first approach is adopted from the trajectory transformer as outlined in (Janner et al., 2021), which organizes the inputs as $(s_1, \ldots, s_t, a_1, \ldots, a_t, r_1, \ldots, r_t)$, grouping states, actions, and rewards accordingly. The second approach is derived from the decision transformer as described in (Chen et al., 2021), and is the method utilized in this study.

---

[1]For better understanding, please refer to https://docs.wandb.ai/guides/app/features/panels/parameter-importance?_gl=1*4s7cuj*_ga*MTQxNjYxODU0OC4xNjgzNjY4Nzg3*_ga_JH1SJHJQXJ*MTY4NDc5NDkzNS40MS4xLjE2ODQ3OTQ5NDIuNTMuMC4w

| Game | Choice one | Choice two (Ours) |
|---|---|---|
| Alien | 211.9 | 239.6 |
| MsPacman | 637.1 | 713.4 |
| Pong | 19.0 | 19.1 |
| SpaceInvaders | 165.7 | 171.2 |
| StarGunner | 620.7 | 709.3 |

Table 8: Ablation studies on different choices of organizing. Each value represents raw scores in Atari games.

From the table above, we observe minor differences between the two sets of inputs. However, the variance in outcomes between the two methodologies is not significant. Therefore, in this paper, we empirically adopt the second approach for our design.

## B.7 Ablation studies with DT

| | DT-Mem (Ave) | DT-Mem FT (Ave) | DT-20M (Ave) |
|---|---|---|---|
| 10k | - | - | 10.1% |
| 20k | - | - | 9.8% |
| 30k | - | - | 15.3% |
| 40k | - | - | 22.6% |
| 50k | 51.0% | 127.4% | 41.8% |
| 100k | - | - | 83.1% |
| 200k | - | - | 120.3% |
| 500k | - | - | 170.7% |

Table 9: Comparison with DT in different fine-tuning datasets

As shown in Table 9, the left-most column represents the size of the dataset used for training. As seen in the table above, the generalized agent DT-Mem outperforms when compared to training on the DT-20M 50k datasets. Fine-tuning DT-Mem on 50k datasets yields better results than training DT-20M on 200k datasets.

## B.8 Full Fine-tuning vs. LoRA

**Full Fine-tuning (FFT) vs. LoRA**: To assess whether the use of LoRA adversely affects performance, we conducted experiments contrasting Full Fine-Tuning (FFT) of memory parameters with LoRA. In this context, FFT-single refers to fine-tuning all parameters exclusively on a single game, whereas FFT-All represents fine-tuning on the entire set of games simultaneously. Results are DQN-normalized score. Based on above results, we conclude the following observations:

| Game | PEFT | FFT-Single | FFT-All |
|---|---|---|---|
| Alien | 127.4% | 116.8% | 113.9% |
| MsPacman | 130.8% | 122.8 | 77.1% |
| Pong | 0% | 0% | 0% |
| SpaceInvaders | 100.8% | 86.8% | 73.4% |
| StarGunner | 158.3% | 55.7% | 40.6% |

- LoRA appears to be the most consistently effective strategy across the games provided. - While **FFT-Single** occasionally outperforms PEFT (like in Alien), **FFT-All** consistently trails behind the other two.

The reason full fine-tuning is not comparable to PEFT comes from the following parts: 1. Fine-tuning dataset size. Note that we only use 50k data in LoRA and full fine-tuning compares on 500k used in MDT paper 2. The benefits of LoRA is: "This approach also addresses catastrophic forgetting and has outperformed standard fine-tuning in low-data and out-of-domain situations"

### B.9  ANALYZE OF INPUT MISLEADING

we conducted an experiment to assess the robustness of the proposed method against input distortion. This involved adding Gaussian noise to the input frames of Atari games. Specifically, we set the mean to 0 and experimented with various standard deviation values. The results are detailed in the table below:

|  | Alien | MsPacman | SpaceInvaders | StarGunner |
|---|---|---|---|---|
| MDT | 3.8% | 13.2% | 8.6% | 2.3% |
| DT-Mem | 51.0% | 69.3% | 53.6% | 62.2% |
| DT-Mem (std=0.5) | 55.3% | 67.6% | 53.0% | 57.8% |
| DT-Mem (std=1) | 35.6% | 56.1% | 40.0% | 34.6% |
| DT-Mem (std=2) | 25.9% | 35.6% | 30.5% | 21.1% |

From the results above, we conclude that the proposed DT-Mem demonstrates greater robustness to noisy inputs compared to the MDT method. This is evident as the DT-Mem consistently outperforms MDT under various levels of Gaussian noise. Notably, the performance with a standard deviation of 0.5 shows minimal difference compared to the no-noise scenario, illustrating DT-Mem's effectiveness in mitigating the impact of varying input distortions.

## C  MEMORY MODULE VISUALIZATION

Figure 10 illustrates the visualization of the memory module. Since memory operations are trained in conjunction with the transformer module, we select a later training episode at random to mitigate uncertainties regarding operational parameters. Due to time constraints, we trained on only two games simultaneously. In the revised version of the paper, we intend to provide visualizations for all games. For clearer visualization, we opted for a memory module of a smaller size, containing 128 memory slots.

Let's first discuss how memory modules update within the same game. As observed in the figure, for the Amidar game, the actively updated memory slots concentrate around rows 18, 84, and 117. This pattern is consistent across episodes, albeit with reduced activity. Such a trend indicates that during each training iteration, the transformer agent tends to overwrite the same memory slot contents. We note a similar observation in the Assault game. Furthermore, we observe that the memory module's activity diminishes in later episodes. For instance, in the Assault game, the active memory slot in row 12 during episode 200k becomes less active by episode 201k. We hypothesize that as training progresses, the accumulated knowledge becomes sufficiently robust for retrieval, reducing the need for updates.

Moving on, when comparing the activity of memory slots across different games, there are intriguing overlaps. For instance, comparing Amidar 200k and Assault 200k reveals that memory slots around row 120 are active in both games. We surmise that this region retains cross-task knowledge shared between games. Additionally, the varying attention across other memory slots demonstrates how these slots assist the agent in decision-making across diverse games.

## D  LIMITATIONS AND SOCIETAL IMPACT

**Limitations** The first limitation of our work is the sample efficiency of memory fine-tuning. The 10% fine-tuning dataset is still sizeable, and we plan to explore more sample-efficient methods in the future. We could, for instance, consider a setting with more tasks, each one with less data, so that the inter-task generalization would be even more crucial to its performance. Additionally, this work does not propose a control strategy for collecting data on a new task. For future work, we plan to investigate online data collection methods, which include the design and learning of exploration strategies for an efficient fine-tuning on new tasks. Finally, the approach has been intuitively motivated, but it would be valuable to have a theoretical grounding that would show the structural limits of large models and how equipping them with a memory component overcomes them.

**Societal Impact** We do not foresee any significant societal impact resulting from our proposed method. The current algorithm is not designed to interact with humans or any realistic environment

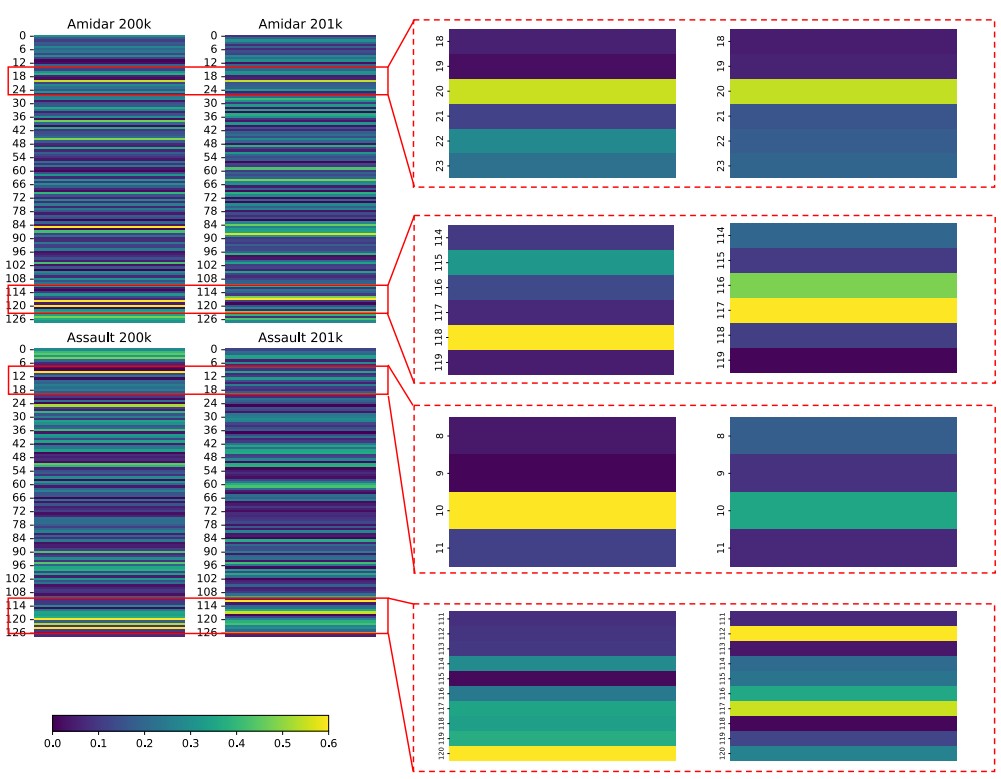

Figure 10: This visualization represents the memory module. In the figure, each row is derived from the mean of a vector that signifies a memory slot. Each depiction calculates the variation between two write operations in a single episode for each memory slot. Lighter shades indicate memory slots that have been actively updated post-write operations. The encircled areas highlight the comparison of active memory slots across different episodes.

yet. If one chooses to extend our methods to such situations, caution should be exercised to ensure that any safety and ethical concerns are appropriately addressed. As our work is categorized in the offline-RL domain, it is feasible to supplement its training with a dataset that aligns with human intents and values. However, one must be wary that the way our architecture generalizes across tasks is still not well understood, and as a consequence, we cannot guarantee the generalization of its desirable features: performance, robustness, fairness, etc. By working towards methods that improve the computational efficiency of large models, we contribute to increasing their access and reducing their ecological impact.

# E    COMPARISON OF DT-MEM AND NEURAL EPISODIC CONTROL (NEC) IN WRITING AND READING MEMORY

## MEMORY MECHANISM

- **NEC:** Utilizes a Differentiable Neural Dictionary (DND) for storing experiences with separate memories for each action, containing state representations (keys) and value function estimates (values).
- **DT-Mem:** Integrates an internal memory module within a transformer framework, focusing on storing, blending, and retrieving information for improving training efficiency and generalization.

## WRITING TO MEMORY

- **NEC:** Continuously adds new experiences and rapidly updates value function estimates in memory.
- **DT-Mem:** Modifies or replaces existing information in the memory matrix using an attention mechanism to calculate correlations and update memory with the attended weight of the input sequence.

## READING FROM MEMORY

- **NEC:** Implements context-based lookups in the DND to retrieve values, outputting a weighted sum based on the similarity between the current state and stored keys.
- **DT-Mem:** Employs content-based addressing for memory retrieval, using attention mechanisms to read from the updated memory and inform decision-making.

## DISTINCTIVE FEATURES AND ADVANTAGES

- **NEC:** Designed for rapid assimilation and action upon new experiences with specialized and swift updates for each action.
- **DT-Mem:** Aims to enhance generalization across tasks and reduce catastrophic forgetting by integrating memory with the transformer's sequential data handling capabilities.