# OpenReview forum: "Think Before You Act: Decision Transformers with Internal Memory"
_ICLR.cc/2024/Conference — Submitted to ICLR 2024_

### Official Review · Reviewer_NgFj · 2023-10-24

**Soundness:** 3 good
**Presentation:** 2 fair
**Contribution:** 2 fair
**Rating:** 5
**Confidence:** 4

**Summary:**

The paper proposes a memory module for Transformer-based architecture that can store and retrieve information for multiple Reinforcement Learning (RL) tasks. Memory update modifies existing information in the memory matrix based on the input sequence and the attention mechanism. Memory retrieval accesses the memory matrix based on content-based addressing. This memory module is integrated with a pre-trained Decision Transformer ((GPT2 architecture) for multi-task RL settings, coupled with a low-rank adaptation fine-tuning method (LoRA). The paper examines the proposed method on multi-game Atari and meta-world object manipulation benchmarks, showing consistent improvements in terms of generalization, adaptation and scaling.

**Strengths:**

- The general motivation is good, a memory module will enhance memorization and can be potentially beneficial for the multi-task RL setting
- The experiments show good results with clear improvement gains

**Weaknesses:**

- The novelty is limited. The main idea is to integrate an external memory for Transformers to improve memorization and reasoning. There have been many works along this line of research.  The proposed memory read and write mechanism is also straightforward and heavily based on the mechanism (content-based attention, add, erase, ...) of NTM and DNC.
- It is unclear why the proposed memory has advantages over other Transformers with internal/external memory or even just long-range attention [1,2,3,4,5]. More explanations are required in the method and more baselines need to be included in the experiments. In particular, the current baselines have only RMDT as a memory-based Transformer, which is not enough, especially when other memory-based Transformers can be adapted easily to offline RL in the same way as the Decision Transformer. Also, retrieval-based RL [6] can be a good baseline as well to highlight the benefit of internal memory.
- The writing lacks sufficient background content. The paper should provide more details on Decision Transformer and offline RL setting.
- Although the main message is about memory, there is no experimental analysis of how the memory module helps improve performance. Please consider ablation studies and visualization to prove that memory is the real contribution (not representation and LoRA tricks). There are some results in Appendix, but they are not helpful (see Questions for more discussion)
- The related work section should include more memory-based Transformer papers

[1] Rae, Jack W., Anna Potapenko, Siddhant M. Jayakumar, and Timothy P. Lillicrap. "Compressive transformers for long-range sequence modelling." arXiv preprint arXiv:1911.05507 (2019).
[2] Martins, Pedro Henrique, Zita Marinho, and André FT Martins. "$\infty $-former: Infinite Memory Transformer." arXiv preprint arXiv:2109.00301 (2021).
[3] Wang, Weizhi, Li Dong, Hao Cheng, Xiaodong Liu, Xifeng Yan, Jianfeng Gao, and Furu Wei. "Augmenting Language Models with Long-Term Memory." arXiv preprint arXiv:2306.07174 (2023).
[4] Wu, Yuhuai, Markus N. Rabe, DeLesley Hutchins, and Christian Szegedy. "Memorizing transformers." arXiv preprint arXiv:2203.08913 (2022).
[5] Wang, Sinong, Belinda Z. Li, Madian Khabsa, Han Fang, and Hao Ma. "Linformer: Self-attention with linear complexity." arXiv preprint arXiv:2006.04768 (2020).
[6] Goyal, Anirudh, Abram Friesen, Andrea Banino, Theophane Weber, Nan Rosemary Ke, Adria Puigdomenech Badia, Arthur Guez et al. "Retrieval-augmented reinforcement learning." In International Conference on Machine Learning, pp. 7740-7765. PMLR, 2022.

**Questions:**

- Are $w$ in Line 202, 215, and 232 the same?
- What is the motivation to compute the strength $\beta$ using attention?  Why do we need to use $\beta$ in both erasing and adding vectors?
- Is $t$ in Line 222 the step in the trajectory? Can you provide an algorithm to explain clearly how memory read and write are executed within a trajectory?
- Is Step 1 Line 187 important? Do you have an ablation study on Step 1?
- Based on Table 5, it seems that LoRA is the main contributor to your method. Can you have the ablation on LoRA using Atari games? Also ablation study on memory adding and erasing would be helpful.
- Can you have visualization to show that the memory stores important data and your model actually reads meaningful memory data from the memory? E.g., when taking action, the model refers to a meaningful timestep in the past to support your idea "think before you act"
- Fig. 3 does your method perform well at 10M parameters?

---

> ### Author Response · Authors · 2023-11-14
>
> Thank you for your insightful feedback!
>
> ## Q1
>
> >The novelty is limited. The main idea is to integrate an external memory for Transformers to improve memorization and reasoning. There have been many works along this line of research. The proposed memory read and write mechanism is also straightforward and heavily based on the mechanism (content-based attention, add, erase, ...) of NTM and DNC.
>
> A: **Internal Memory vs. NTM**: While both the internal memory module and NTM aim to introduce memory mechanisms to neural architectures, our internal memory is inspired by human cognitive processes and is specifically tailored to decision-making agents. The design and integration of this module into DT, and its subsequent application, is a **novel contribution** in the context of our work. Besides, there are other papers inspired by NTM: \[1\] Memory Augmented Neural Networks with Wormhole Connections. ArXiv:1701.08718 \[2\] Hybrid computing using a neural network with dynamic external memory. Nature 2016. Same as the above mentioned papers, despite that our work inspired by NTM, we design a new memory reading and writing techniques for DT.
>
> **Use of LoRA**: While the concept of LoRA might not be new, its application in conjunction with our internal memory module and within the realm of LLM-based decision-making agents brings a novel perspective. The combination of these components, and the synergies they produce, offers a unique approach to addressing the generalization and adaptation in MDT.
>
> The novelty of our work isn't merely in the individual components but in the holistic approach and the results achieved. By blending inspiration from human cognition with state-of-the-art machine learning techniques, our paper is the **first** paper that introduces internal memory in DT improved training efficiency and generalization in a variety of tasks, as evidenced in our experiments.
>
> **Contextual Novelty**: It's crucial to recognize that novelty doesn't always stem from introducing entirely new concepts. Often, the innovative combination, adaptation, and application of existing ideas to new contexts or challenges can bring significant advancements to a field.

---

> > ### Author Response · Authors · 2023-11-14
> >
> > ## Q2
> >
> > >It is unclear why the proposed memory has advantages over other Transformers with internal/external memory or even just long-range attention [1,2,3,4,5]. More explanations are required in the method and more baselines need to be included in the experiments. In particular, the current baselines have only RMDT as a memory-based Transformer, which is not enough, especially when other memory-based Transformers can be adapted easily to offline RL in the same way as the Decision Transformer. Also, retrieval-based RL [6] can be a good baseline as well to highlight the benefit of internal memory.
> >
> > A: Thank you for the valuable suggestions. However, RMDT is the most state-of-the-art method that related to our settings, memory module for DT.
> >
> > **Compare to existing methods**:
> >
> > - **Memory Writing**:
> > 	1. **∞-former** represents memory as a continuous signal using radial basis functions (RBFs). When new information is encountered, it is integrated into this continuous representation. This process involves evaluating the continuous signal at specific locations and then concatenating these evaluations with new vectors coming from the short-term memory.
> > 	2. **DT-Mem** involves sophisticated mechanisms using attention to manage the significance of new and existing data. This process would be based on calculating correlations between the input and memory and updating the memory with the attended weight of the input sequence.
> > 	3. **LONGMEM** caches paired attention keys and values from the previous context into a non-differentiable memory bank using a frozen backbone Large Language Model (LLM) as the memory encoder.
> > 	4. **KNN-transformer** uses (key, value) pairs from the local context are appended to the end of an external memory.
> > - **Memory Reading**:
> > 	1. The reading process of **∞-former** utilizes a continuous-space attention framework.
> > 	2. **DT-Mem** uses content-based addressing for memory retrieval. This process would involve using attention mechanisms to read from the updated memory, focusing on the content relevant to the current task or context.
> > 	3. **LONGMEM** uses a decoupled memory module, specifically a residual side-network (SideNet), for memory retrieval and reading. The SideNet retrieves cached key-value pairs of previous contexts from memory using the attention query of the current input.
> > 	4. **KNN-transformer**: features a kNN-augmented attention layer that combines standard dense self-attention with approximate k-nearest-neighbor search into the external memory. The kNN lookup retrieves the top-k (key, value) pairs for each query from the input subsequence, constructing an attention matrix that represents these memories differently for each query.
> >
> > In summary:
> >
> > - **DT-Mem**: Stands out for its efficient decision-making capabilities and balanced approach to memory retention and adaptation, making it particularly suitable for tasks requiring contextually relevant decisions.
> > - **∞-former**: Excels in handling unbounded contexts with precision and flexibility, ideal for extensive and complex datasets.
> > - **LONGMEM**: Its decoupled architecture and token-to-chunk retrieval make it adept at managing large contexts.
> > - **kNN-augmented Transformer**: Offers flexibility in context length and rapid adaptation to new data, enhancing the model's real-time applicability.
> >
> > Linformer is a method trying to reduce the complexity of transformer model, which is not related to our paper. Thus, we don’t compare with this method.
> >
> > Due to the limit time, we would love it if the reviewer could provide a bit more information about what are the experiments they would like us to prioritize, what they hope to get from the additional results they ask of us, and how that would affect their assessment of our work.
> >
> >
> > ## Q3
> >
> > >The writing lacks sufficient background content. The paper should provide more details on Decision Transformer and offline RL setting.
> >
> > A: Thank you for the comment. We have added more details about DT and offline RL in the paper revision Section 3.1 lines 141-146.
> > ## Q4
> >
> > >The related work section should include more memory-based Transformer papers
> >
> > A: Thank you for the suggestion. We have added these related works in the paper revision lines 121-125.

---

> ### Author Response · Authors · 2023-11-14
>
> ## Q5
>
> >Are $\omega$  in Line 202, 215, and 232 the same?
>
> A: Yes, the address $\omega$ remains the same in these lines. Line 202 defines how we compute the address. Line 215 defines how we write information to the memory, and line 232 defines how we read from the memory.
>
> The reasons are as follows:
>
> 1. **Content-based Addressing**: DT-Mem uses a content-based addressing mechanism, where the content to be read or written influences the memory address. Since we are calculated the address using current input embeddings $E$, this mechanism aligns with using the same address for both operations, as the content being processed is directly tied to where it is stored or retrieved from.
> 2. **Memory Consistency and Coherence**: This approach allows the model to immediately read back what it has just written, ensuring that the memory reflects the latest changes made by the network.
> 3. **Simplified Memory Management**: Using the same address for reading and writing simplifies the memory management process. It reduces the complexity of the controller's operations, as it doesn't need to maintain separate mechanisms or algorithms for addressing reads and writes.
>
> ## Q6
>
> >What is the motivation to compute the strength $\beta$ using attention? Why do we need to use $\beta$ in both erasing and adding vectors?
>
> A: The primary motivation behind employing attention mechanisms is to achieve **Context-Awareness**. By using cross-attention between the input embeddings, $E$, and the memory, $M$, the model can selectively concentrate on pertinent aspects of the input. This attention-based computation ensures that memory updates are context-dependent, enhancing both the dynamism and efficiency of the process.
>
> The computation of $\beta$ plays a crucial role in regulating the flow of information to the memory. As $\beta$ represents a balance between the information in the addition vector $ \epsilon^a$ and the erasure vector $ \epsilon^e$, it effectively governs the extent of new information added to, and old information removed from, the current memory. This parameter allows for dynamic control over memory updates. For instance, when the input information is highly relevant to specific memory segments of interest (as determined by the content-address $\omega $), $\beta$ assumes a larger value. Consequently, only a small portion of memory is erased (calculated as 1-$\beta$, as outlined in line 216) and predominantly mixed with the addition vector (as described in line 222). Conversely, if the input information significantly differs, the model erases more from the current memory, adding only a minimal amount of new information.
>
>
> ## Q7
>
> >Is $t$  in Line 222 the step in the trajectory? Can you provide an algorithm to explain clearly how memory read and write are executed within a trajectory?
>
> A: Thank you for highlighting this point of confusion. In this context, the symbol $t$ represents the $N$th timestep of memory operations that occur within the interactions between the transformer module and memory. It specifically refers to the internal loop of memory operations and is not associated with the trajectory. For a detailed explanation, we have included the relevant algorithm in Appendix Algorithm 3.
>
>
> ## Q8
>
> >Is Step 1 Line 187 important? Do you have an ablation study on Step 1?
>
> A: Step 1 serves two main purposes:
>
> 1. To map various input dimensions into a consistent dimension for storage. Given that our memory is a matrix with fixed dimensions (i.e., number_of_slots * dimensions), it's imperative to harmonize input dimensions for storage.
> 2. As highlighted in our paper, the input sequence comprises multiple steps of a tuple $< \hat{r}_t, s_t, a_t >$. We use the timestep $t$ to arrange these inputs, thereby preserving the temporal relationships among them.
>
> We've conducted results without including input sequence organization:
>
> |Game|DT-Mem (No-step 1)|DT-Mem|
> |---|---|---|
> |Alien|211.9|239.6|
> |MsPacman|637.1|713.4|
> |Pong|19.0|19.1|
> |SpaceInvaders|165.7|171.2|
> |StarGunner|620.7|709.3|
>
> Though minor differences exist between the two sets of inputs, the variance in outcomes between the two methodologies is not pronounced.

---

> > ### Author Response · Authors · 2023-11-14
> >
> > ## Q9
> >
> > >Based on Table 5, it seems that LoRA is the main contributor to your method. Can you have the ablation on LoRA using Atari games? Also ablation study on memory adding and erasing would be helpful.
> >
> > A: We argue this findings since we **never** claim LoRA as a main contributor to the method. Instead, **memory module is the main contributor** to our method, as illustrated in experiments Table 1, 2, Figure 3,4,5,6. These results all denote the effectiveness of proposed memory module in both generalization and adaptation scenarios.
> >
> > The LoRA is a method we used to fine-tune the memory module, this method can be replaced by other fine-tuning methods and we witness the similar phenomenon that memory module increase the model adaptability. The results can be find at the end.
> >
> > The evaluation results show in Section 5.5 clearly denotes the ablation on LoRA using Atari games.
> >
> > **Full Fine-tuning (FFT) vs. LoRA**: To assess whether the use of LoRA adversely affects performance, we conducted experiments contrasting Full Fine-Tuning (FFT) of memory parameters with LoRA. In this context, FFT-single refers to fine-tuning all parameters exclusively on a single game, whereas FFT-All represents fine-tuning on the entire set of games simultaneously. Results are DQN-normalized score.
> >
> > |Game|LoRA|FFT-Single|FFT-All|
> > |---|---|---|---|
> > |Alien|127.4%|116.8%|113.9%|
> > |MsPacman|130.8%|122.8|77.1%|
> > |SpaceInvaders|100.8%|86.8%|73.4%|
> > |StarGunner|158.3%|55.7%|40.6%|
> >
> > Based on above results, we conclude the following observations:
> >
> > - LoRA appears to be the most consistently effective strategy across the games provided.
> > - While **FFT-Single** occasionally outperforms PEFT (like in Alien), **FFT-All** consistently trails behind the other two.
> >
> > The reason full fine-tuning is not comparable to PEFT comes from the following parts: 1. Fine-tuning dataset size. Note that we only use 50k data in LoRA and full fine-tuning compares on 500k used in MDT paper 2. The benefits of LoRA listed above: "This approach also addresses catastrophic forgetting [2] and has outperformed standard fine-tuning in low-data and out-of-domain situations [3]”
> >
> > [1] LoRA: Low-Rank Adaptation of Large Language Models, ICLR 2022
> >
> > [2] Adapter-Fusion: Non-destructive task composition for transfer learning.EACL, 2021.
> >
> > [3] Prefix-tuning: Optimizing continuous prompts for generation. ACL, 2021.
> >
> >
> > ## Q10
> >
> > > Can you have visualization to show that the memory stores important data and your model actually reads meaningful memory data from the memory? E.g., when taking action, the model refers to a meaningful timestep in the past to support your idea "think before you act"
> >
> > A: Please refer to Appendix Section C for visualization on memory module
> >
> >
> > ## Q11
> >
> > >Fig. 3 does your method perform well at 10M parameters?
> >
> > A: Our proposed method, DT-Mem, demonstrates limited performance at a model size of 10M parameters, achieving only a 26% human-normalized IQM score. This underperformance can be attributed to the additional parameter requirements for memory management and calculations, as detailed in Appendix Table 3. With a total parameter count of 10M, only about 3M parameters are allocated to the transformer and MLP modules. We believe that this allocation is insufficient for achieving optimal performance, as these modules require a larger parameter share to function effectively.

---

> > > ### Author Response · Authors · 2023-11-22
> > > **Thank you for the review and awaiting your response**
> > >
> > > As the end of the author-reviewer discussion is approaching, we kindly ask for your review of our revised paper and response. If our revisions have adequately addressed your concerns, we'd greatly appreciate your consideration in adjusting the scores. Should you have any remaining questions or require further clarification, please don't hesitate to let us know. We would be more than happy to answer any further questions.

---

### Official Review · Reviewer_qzjQ · 2023-10-31

**Soundness:** 2 fair
**Presentation:** 2 fair
**Contribution:** 2 fair
**Rating:** 6
**Confidence:** 4

**Summary:**

In this paper, the authors introduce a novel agent algorithm called Decision Transformer with Memory (DT-Mem), which incorporates a memory layer between the attention and MLP layers in the Transformer architecture. This addition allows the agent to memorize knowledge in memory, rather than relying solely on learnable parameters. Empirical evidence demonstrates that DT-Mem outperforms existing methods in terms of generalization and can quickly adapt to new tasks through fine-tuning, such as LoRA.

**Strengths:**

- The paper presents the Decision Transformer with Memory (DT-Mem) model, which incorporates a learnable memory module and demonstrates superior performance in pre-training, generalization, and fine-tuning. It distinguishes itself from related work, such as RMDT, by its ability to learn sequences in parallel training and use an advanced learnable memory architecture based on the NTM model.
- In section 4, the paper's clarity for the proposed model is commendable, with a well-structured architecture diagram and detailed explanations for each step of inference and training.
- The evaluation methodology includes several well-designed questions that effectively assess the hypotheses presented in the paper.
- Comparative evaluation against diverse baselines, including another memory-equipped Decision Transformer model (RMDT), strengthens the paper's contributions.
- In Figures 3 and 4, comparison with diverse size of MDT is interesting, through this, we can clearly see that the memorization through the neural network parameters is inefficient compared to the implicit learnable memory.
- The additional experimental results in the appendix are helpful to understand deeply (especially Figure 10 is good).

**Weaknesses:**

- Some explanations in the paper are unclear. For instance, the reference to "Large Language Model based decision making agents" needs clarification. It's unclear if this refers to Transformer-based agents or models within the Decision Transformer family.
- The motivation for this work is not entirely clear. While the paper argues that memorization through neural network parameters can lead to weaker performance in forgetting during training, it remains unclear how this relates to generalization performance for unseen tasks.
- The paper contains some comments that are difficult to understand, such as the mention of NFT and FT in Figure 5, where no results are presented for these abbreviations.
- There's a minor typographical error in line 297, where "we generat" should be corrected to "we generate."

**Questions:**

- Is the memory not initialized throughout the entire training process? Clarifying this point could help readers better understand the novelty of this approach, as memory initialization is typically performed per episode (e.g., NTM).
- Could you investigate whether MDT can be fine-tuned using the LoRA technique? As LoRA is applicable to the general Transformer architecture, it would be insightful to assess the potential for fine-tuning MDT using this approach.
- Have you considered testing an external memory-equipped DT? Many recent attempts to incorporate memory utilize a naive appending-style external memory, which lacks the sophistication of models like NTM. Comparing your memory architecture to this version could help highlight its strengths.
- Expanding the tasks to include those with long-term dependencies, such as MemoryMaze, could be valuable. Given that DT-Mem has a memory module capable of retaining distant knowledge, it may excel in tasks where other DTs, except RMDT, struggle.
- In line 243, the phrase "the Transformer module to generate actions that can reduce this value as close to zero as possible" may benefit from clarification. The objective appears to be maximization rather than reduction, as it involves the sum of rewards.
- Figure 5 raises questions about the absence of NFT results. Are all results in the plot derived from fine-tuned models?
- In Figure 6, where you mention the top 3 rollout, could you provide more context or clarification about what this entails?

### Additional Comments
DT-Mem demonstrates superior performance in pre-training, generalization, and fine-tuning, particularly evident in Figure 10. However, there seems to be a disconnect between the paper's motivation and the observed results. Clarifying the link between implicit memory's ability to mitigate the forgetting phenomenon and the improved generalization and fine-tuning performance would strengthen the paper's alignment and overall impact.

### After reading the author's rebuttal
We thank the authors for their efforts to clarify their arguments. I agree with their rebuttal, in particular, the part related to the connection between their motivation and their methodologies. I hope they will try to test the appending-style external memory also, but I am satisfied to their rebuttal, so I increase my score to lean to the acceptance.

To authors, as I know, there is no prior work for the appending-style external memory + DT, but the external memory equipped agents have been studied actively. I leave some references.

Lampinen, Andrew, et al. "Towards mental time travel: a hierarchical memory for reinforcement learning agents." Advances in Neural Information Processing Systems 34 (2021): 28182-28195.

Parisotto, Emilio, et al. "Stabilizing transformers for reinforcement learning." International conference on machine learning. PMLR, 2020.

---

> ### Author Response · Authors · 2023-11-14
>
> Thank you for your insightful feedback!
>
> ## Q1
>
> > Some explanations in the paper are unclear. For instance, the reference to "Large Language Model based decision making agents" needs clarification. It's unclear if this refers to Transformer-based agents or models within the Decision Transformer family.
>
> A: This phrase generally refers to all the large language model-based agents, which includes both decision transformer, trajectory transformer [1] and the following works. In this paper, we mainly focus on decision transformer family. We’ve revised this sentence in the paper revision line 1.
>
> [1] Offline Reinforcement Learning as One Big Sequence Modeling Problem. [https://arxiv.org/abs/2106.02039v4](https://arxiv.org/abs/2106.02039v4)
>
>
> ## Q2
>
> >The motivation for this work is not entirely clear. While the paper argues that memorization through neural network parameters can lead to weaker performance in forgetting during training, it remains unclear how this relates to generalization performance for unseen tasks.
>
> A: **Memorization and Forgetting in Neural Networks**: When a neural network is trained on a series of tasks, it tends to "memorize" the specifics of those tasks through its parameters. This process can lead to a phenomenon known as "catastrophic forgetting," where learning new tasks impairs the network's performance on previously learned tasks [1]. This is a significant issue in the context of decision-making models, where adaptability and the ability to handle a variety of scenarios are crucial.
>
> **Generalization to Unseen Tasks**: Generalization refers to a model's ability to perform well on new, unseen tasks, not just the ones it was trained on. The problem with the traditional approach of memorizing through parameters is that it often leads to overfitting on the training data, which can diminish the model's ability to generalize. When a model is overfitted, it becomes too tailored to the specifics of its training data and loses the flexibility needed to adapt to new situations or tasks [2].
>
> **Relation Between Forgetting and Generalization**: The link between the forgetting phenomenon and generalization is that both are impacted by how a model learns and stores information. A model that is prone to forgetting may struggle with generalization because it cannot effectively retain and utilize the broad range of knowledge required to handle new tasks [3]. Conversely, a model designed to minimize forgetting (such as through an internal memory mechanism like DT-Mem) can potentially maintain a more diverse and generalizable knowledge base.
>
> **Motivation of DT-Mem**: The motivation for introducing DT-Mem, as such, is to address these interrelated issues. By incorporating an internal memory module, DT-Mem aims to reduce the reliance on parameter-based memorization, thereby mitigating the effects of catastrophic forgetting. This is hypothesized to improve the model's ability to generalize to new tasks, as it can more effectively retain and apply knowledge from a broader range of experiences. Section 5.3 Table 1 the results of model generalization support this findings.
>
> [1] Overcoming catastrophic forgetting in neural networks. National academy of sciences 2017.
>
> [2] Understanding deep learning requires rethinking generalization. arXiv preprint arXiv:1611.03530.
>
> [3] Overcoming catastrophic forgetting in neural networks. Proceedings of the national academy of sciences, 114(13), 3521-3526. 2017
>
>
> ## Q3
>
> > Is the memory not initialized throughout the entire training process? Clarifying this point could help readers better understand the novelty of this approach, as memory initialization is typically performed per episode (e.g., NTM).
>
> A: The memory is initialized only once during the entire training process. This approach ensures that the knowledge acquired from one task can be leveraged in subsequent tasks, leading to improved pre-training performance, as evidenced in Fig. 6.
>
>
> ## Q4
>
> > Could you investigate whether MDT can be fine-tuned using the LoRA technique? As LoRA is applicable to the general Transformer architecture, it would be insightful to assess the potential for fine-tuning MDT using this approach.
>
> A: The results shown in Section 5.5 Figure 5 are all fine-tuned using the LoRA technique. As we concluded in lines 342-351. The results show that “the consistent superior performance of DT-Mem across most games suggests that this method might have a more adaptable approach.”

---

> > ### Author Response · Authors · 2023-11-14
> >
> > ## Q5
> >
> > > Have you considered testing an external memory-equipped DT? Many recent attempts to incorporate memory utilize a naive appending-style external memory, which lacks the sophistication of models like NTM. Comparing your memory architecture to this version could help highlight its strengths.
> >
> > A: Thank you for the valuable suggestions. However, we couldn’t find any papers related to external memory-equipped DT. We would love it if the reviewer could provide a bit more information about what are the experiments they would like us to prioritize, what they hope to get from the additional results they ask of us, and how that would affect their assessment of our work.
> >
> >
> > ## Q6
> >
> > > Expanding the tasks to include those with long-term dependencies, such as MemoryMaze, could be valuable. Given that DT-Mem has a memory module capable of retaining distant knowledge, it may excel in tasks where other DTs, except RMDT, struggle.
> >
> > A: We greatly appreciate your suggestion. We are actively conducting the recommended experiments and will promptly share the results as soon as they become available.
> >
> >
> > ## Q7
> >
> > >In line 243, the phrase "the Transformer module to generate actions that can reduce this value as close to zero as possible" may benefit from clarification. The objective appears to be maximization rather than reduction, as it involves the sum of rewards.
> >
> > A: Thank you for highlighting the confusion regarding this section. We adhere to the conventional 'return-to-go' approach in Decision Transformers, where the return-to-go at time step $t$, denoted as $\hat{r}_t$, represents the total expected reward from that point to the episode's end. The training objective for models like Decision Transformers typically focuses on maximizing the expected return, or equivalently, minimizing the negative return. In our case, the goal is specifically to minimize the **negative return** as much as possible, aiming towards zero. This clarification has been incorporated in the revised manuscript at line 246.
> >
> >
> > ## Q8
> >
> > >Figure 5 raises questions about the absence of NFT results. Are all results in the plot derived from fine-tuned models? In Figure 6, where you mention the top 3 rollout, could you provide more context or clarification about what this entails?
> >
> > A: Yes, all results in the plot derived from fine-tuned models. The terms 'NFT' and 'top3' were included erroneously and are, as you correctly noted, typographical errors. Thank you for pointing out these issues, we have revised Figure 5 and Figure 6 captions in the paper revision.

---

> ### Author Response · Authors · 2023-11-20
> **Results about MemoryMaze Environment**
>
> We evaluated both DT-Mem and MDT using the Memory Maze offline probing benchmark to determine if DT-Mem enhances performance in tasks requiring long-term dependency. Performance was measured using wall prediction accuracy (%) for the Walls benchmarks (higher the better) and mean-squared error for the Objects benchmarks (lower the better). We report the average score across three training runs. The results are as follows:
>
> |  | Memory 9x9 Walls | Memory 15x15 Walls | Memory 9x9 Objects | Memory 15x15 Objects |
> | --- | --- | --- | --- | --- |
> | Constant Baseline | 80.8% | 78.3% | 23.9 | 64.8 |
> | MDT | 87.2%$\pm 0.4$ | 79.3%$\pm 0.7$ | 8.6$\pm 0.3$ | 32.2$\pm 0.2$ |
> | DT-Mem | 96.9%$\pm 0.3$ | 80.6%$\pm 0.5$ | 3.0$\pm 0.3$ | 27.8$\pm 0.3$ |
>
> These results indicate that DT-Mem achieves superior performance in both environments across all four metrics. This suggests that the incorporation of a memory module in DT-Mem effectively stores past information for future decision-making. It highlights the method's efficacy and its advantages in handling long-term dependency tasks.

---

> > ### Comment · Reviewer_qzjQ · 2023-11-21
> > **Response to the rebuttal**
> >
> > We thank the authors for their efforts to clarify our concerns. They addressed our concerns properly, and we agree with their explanation about the connection between their motivation and methodology. Thus, we increased our score to lean to the acceptance. We leave the additional comments in our review, please check that.

---

> > > ### Author Response · Authors · 2023-11-21
> > >
> > > Thank you for adjusting the score and for your additional insightful comments. We will add these related works in the paper revision.

---

### Official Review · Reviewer_hmxu · 2023-11-01

**Soundness:** 3 good
**Presentation:** 3 good
**Contribution:** 2 fair
**Rating:** 6
**Confidence:** 4

**Summary:**

This paper introduces the memory into the decision transformer. Basically, the memory is a matrix which store the embedding of the transition tuple, and when decision making using decision transformer, the retrieved information from the memory is used to generate the action. Experiments on Atari and Meta-world demonstrate the effectiveness of the proposed methods.

**Strengths:**

The motivation of this paper is clear, i.e., inspiring by the human decision making process.
The writing of the paper is good, i.e., easy to follow.

**Weaknesses:**

The main contribution of the paper is the memory. however, more in-depth investigation of the memory can be conducted. There are some issues about the clarity.

**Questions:**

There are several questions I want the author to address during the rebuttal:
1. Some issues about the clarity:
a. For Section 3.1. It seems that you focus on offline RL, rather than RL. The two fields have differences. You may need a brief introduction of offline RL, as well as existing methods as baselines, such as DT, MGDT, RMDT, HDT.
b. Figure 2 is somehow misleading. Figure 2a, does the memory module is a layer of the transformer? I think they are separated, however, the plot seems that the memory is stacked with the transformer. Figure 2b, the retrieved memory is another memory? I think should be the retrieved experiences, or other terms. Please make the terms unique and clear.

2. Some issues about the technical contributions.
a. The introduction uses large language model as the motivation, however, even using GPT-2 architecture, the embedding and the tokens I believe is not about words, it should be game-specific tokens. so please using transformer or decoder-only transformer to avoid any confusion.
b. It seems that the largest model used in the paper is 50M. Compared with LLM, it still very small. Does LoRA is necessary? LoRA can be used to any models, which is not a technical contribution of this paper. However, that may harm the performance of DT-Mem. So I would suggest just not using LoRA and fine-tuning all the model to focus on the memory part. This can help the reviewer to fully evaluate the importance of the memory.
c. About the memory. There are some related methods, neural episodic control (https://arxiv.org/abs/1703.01988) for the writing and lookup. The two methods share many similarities, so maybe add a detailed comparison of the proposed methods and all related methods, so we can fully understand the contributions.
d. Still about the memory.  What is exactly the difference between the external memory and the internal memory? Could you provide an example about the two kinds of memories, as well as the advantages of the proposed memory.

3. Some issues about the experiments. I generally think the experiments are sufficient, but with some suggestions: i) can we use the prompting for the DT-Mem? as we know prompting is much easier than fine-tuning. And ii) what is the limit of the internal memory, given the fixed size, i.e., parameters, of the memory?

---

> ### Author Response · Authors · 2023-11-14
>
> Thank you for your insightful feedback!
>
> ## Q1.a
>
> >For Section 3.1. It seems that you focus on offline RL, rather than RL. The two fields have differences. You may need a brief introduction of offline RL, as well as existing methods as baselines, such as DT, MGDT, RMDT, HDT.
>
> A: We appreciate the insightful suggestion. Accordingly, a concise overview of offline RL has been incorporated into Section 3.1 (lines 141-146). Additionally, due to page constraints, we have briefly introduced the baselines in Section 5.2.
>
>
> ## Q1.b
>
> > Figure 2 is somehow misleading. Figure 2a, does the memory module is a layer of the transformer? I think they are separated, however, the plot seems that the memory is stacked with the transformer. Figure 2b, the retrieved memory is another memory? I think should be the retrieved experiences, or other terms. Please make the terms unique and clear.
>
> A: We appreciate the suggestion. To clarify, in Figure 2a, the memory module is depicted as an independent module, distinct from the transformer module. Conversely, in Figure 2b, the memory that is retrieved serves as an intermediate input to the transformer module, integrating with it more directly. We have updated Figure 2 in the revised version of the paper to better illustrate these configurations and their respective operational dynamics.
>
>
> ## Q2.a
>
> > The introduction uses large language model as the motivation, however, even using GPT-2 architecture, the embedding and the tokens I believe is not about words, it should be game-specific tokens. so please using transformer or decoder-only transformer to avoid any confusion.
>
> A: Thank you for the comments. We have revised this term in line 16 and line 20.
>
>
> ## Q2.b
>
> >It seems that the largest model used in the paper is 50M. Compared with LLM, it still very small. Does LoRA is necessary? LoRA can be used to any models, which is not a technical contribution of this paper...
>
> A: **Model Size:** The choice of a 50M model size was dictated by the computational resources available to us. Nonetheless, in Figure 3, we explore the scaling law of DT-Mem, which reveals an interesting trend: as the number of parameters increases, so does performance. Importantly, when compared to the 200M MDT model, our DT-Mem, with only a quarter of the parameter size, demonstrates superior performance. This not only underscores the efficiency of DT-Mem in generalizing across games but also empirically supports its effectiveness even with fewer parameters.
>
> **Motivation of LoRA**: The motivations of using LoRA to fine-tune the model can be concluded in following two reasons:
>
> 1. According to a paper \[1\], compared to other adapter methods, LoRA method not only don't introduce inferences latency nor reduce input sequence length while retaining high model quality. It allows for quick task-switching when deployed as a service by sharing the vast majority of the model parameters
> 2. Parameter-efficient fine-tuning (PEFT) approaches (such as LoRA) only fine-tune a small number of (extra) model parameters while freezing most parameters of the pretrained LLMs, thereby greatly decreasing the computational and storage costs \[1\]. This also overcomes the issues of catastrophic forgetting \[2\]. PEFT approaches have also shown to be better than fine-tuning in the low-data regimes and generalize better to out-of-domain scenarios \[3\].
>
> **Full Fine-tuning (FFT) vs. LoRA**: To assess whether the use of LoRA adversely affects performance, we conducted experiments contrasting Full Fine-Tuning (FFT) of memory parameters with LoRA. In this context, FFT-single refers to fine-tuning all parameters exclusively on a single game, whereas FFT-All represents fine-tuning on the entire set of games simultaneously. Results are DQN-normalized score.
>
> |Game|LoRA|FFT-Single|FFT-All|
> |---|---|---|---|
> |Alien|127.4%|116.8%|113.9%|
> |MsPacman|130.8%|122.8|77.1%|
> |SpaceInvaders|100.8%|86.8%|73.4%|
> |StarGunner|158.3%|55.7%|40.6%|
>
> Based on above results, we conclude the following observations:
>
> - LoRA appears to be the most consistently effective strategy across the games provided.
> - While **FFT-Single** occasionally outperforms PEFT (like in Alien), **FFT-All** consistently trails behind the other two.
>
> The reason full fine-tuning is not comparable to PEFT comes from the following parts: 1. Fine-tuning dataset size. Note that we only use 50k data in LoRA and full fine-tuning compares on 500k used in MDT paper 2. The benefits of LoRA listed above: "This approach also addresses catastrophic forgetting [2] and has outperformed standard fine-tuning in low-data and out-of-domain situations [3]”
>
> [1] LoRA: Low-Rank Adaptation of Large Language Models, ICLR 2022
> [2] Adapter-Fusion: Non-destructive task composition for transfer learning.EACL, 2021.
> [3] Prefix-tuning: Optimizing continuous prompts for generation. ACL, 2021.

---

> ### Author Response · Authors · 2023-11-14
>
> ## Q2.c
>
> >About the memory. There are some related methods, neural episodic control ([https://arxiv.org/abs/1703.01988](https://arxiv.org/abs/1703.01988)) for the writing and lookup. The two methods share many similarities, so maybe add a detailed comparison of the proposed methods and all related methods, so we can fully understand the contributions.
>
> A: Thank you for the comments. Due to the page limit, we have added these comparisons in the paper revision Appendix Section E
>
>
> ## Q2.d
>
> >Still about the memory. What is exactly the difference between the external memory and the internal memory? Could you provide an example about the two kinds of memories, as well as the advantages of the proposed memory.
>
> ### External Memory vs. Internal Memory
>
> 1. **External Memory**:
> 	- **Definition**: External memory in AI and machine learning models refers to a separate storage unit that is not part of the core neural network architecture. It's like an additional database or repository that the model can access.
> 	- **Example**: A neural Turing machine (NTM) or Differentiable Neural Computer (DNC), which uses an external memory matrix that it can read from and write to, separate from its neural network parameters.
> 	- **Advantages**:
> 		- **Scalability**: Can store large amounts of data beyond the capacity of the model's parameters.
> 		- **Flexibility**: Allows the model to store and retrieve information in a more structured way, similar to how a computer uses RAM and disk storage.
> 2. **Internal Memory**:
> 	- **Definition**: Internal memory refers to the capacity of a neural network to store information within its own architecture, such as in its weights and activations. It does not use a separate storage unit but relies on the network's inherent structure.
> 	- **Example**: LSTM (Long Short-Term Memory) networks that use their recurrent connections to store information over time within the network.
> 	- **Advantages**:
> 		- **Efficiency**: More efficient in terms of retrieval speed, as the information is stored within the network.
> 		- **Integration**: Better integrated with the model's learning process, enabling more seamless information flow and processing.
>
> ### Advantages of the Proposed Memory
>
> 1. **Contextual Retrieval**: Utilizing mechanisms like attention to retrieve and store information more contextually and relevantly.
> 2. **Enhanced Generalization**: By having a more flexible memory system, the model generalize better to new, unseen tasks.
>
> ## Q3.i
>
> > Some issues about the experiments. I generally think the experiments are sufficient, but with some suggestions: i) can we use the prompting for the DT-Mem? as we know prompting is much easier than fine-tuning.
>
> A: Thank you for the suggestion. The primary objective of the fine-tuning in Section 5.5 is to investigate whether modifications to the memory module can boost the performance of DT-Mem without altering other transformer parameters. This is intended to assess the role of the memory module's stored information in aiding decision-making for new tasks. Since the memory module engages in read and write operations through cross-attention, the use of prompt tuning, which does not alter DT-Mem's stored information, is not aligned with our approach.
>
> Nevertheless, we have conducted a comparative analysis of prompt-tuning DT (PDT) and DT-Mem in Table 2. The results demonstrate the effectiveness of our proposed method and highlight the advantages of fine-tuning the memory module.
>
>
> ## Q3.ii
>
> > And ii) what is the limit of the internal memory, given the fixed size, i.e., parameters, of the memory?
>
> A: The major limitation of internal memory is **Capacity Limitation**: The capacity of internal memory is inherently limited by the size of the neural network, i.e., the number of its parameters and the architecture's complexity. Once these parameters are set, the capacity to store and recall information is fixed. The ability to store information depends on how densely information can be encoded within the network's parameters, which has practical limits.

---

> > ### Comment · Reviewer_hmxu · 2023-11-21
> > **Thanks for the response**
> >
> > Thanks for the response. I think that generally address my concerns and I would keep my score.

---

> > > ### Author Response · Authors · 2023-11-21
> > >
> > > Thank you for your valuable comments to enhance the quality of our work.

---

### Official Review · Reviewer_gRQQ · 2023-11-08

**Soundness:** 3 good
**Presentation:** 2 fair
**Contribution:** 3 good
**Rating:** 6
**Confidence:** 3

**Summary:**

This paper proposes Decision Transformers with Memory, which introduces internal memory mechanism into RL field and improves training efficiency and generalization in both Atari games and meta-world object manipulation tasks.

**Strengths:**

- The experiments are sufficient and the results prove the superiority of this method.
- The paper is well written.

**Weaknesses:**

- The internal memory formulation and some specific details, such as content-based addressing, seems a bit incremental from previous work, the authors should explain the difference more clearly in the method section.

**Questions:**

1. Fig. 2(b) is too simple, making it difficult to correspond one-to-one with the steps in the method part. The authors should make the figure more comprehensive and understandable.
2. There are many papers demonstrating the ideas about internal memory, and the authors should explain the differences with similar methods in more detail in the method section.
3. Add more analysis about different situations, such as the input misleading by the content stored in the memory (i.e., noise or dissimilar pattern), how does the method eliminates this type of impact.

---

> ### Author Response · Authors · 2023-11-14
>
> Thank you for your insightful feedback!
> ## Q1
>
> > Fig. 2(b) is too simple, making it difficult to correspond one-to-one with the steps in the method part. The authors should make the figure more comprehensive and understandable.
>
> A: Thank you for the comments. We have revised this figure in the paper revision Fig. 2
>
>
> ## Q2
>
> > There are many papers demonstrating the ideas about internal memory, and the authors should explain the differences with similar methods in more detail in the method section.
>
>
> A: Thank you for pointing this out. To the best of my knowledge, the work most closely related to ours (DT with internal memory) is RMDT, which we have discussed extensively in the paper. I would greatly appreciate it if the reviewer could share specific papers they believe should be compared to our work. We are more than willing to discuss these differences in the revised version of the paper.
>
>
> ## Q3
>
> > Add more analysis about different situations, such as the input misleading by the content stored in the memory (i.e., noise or dissimilar pattern), how does the method eliminates this type of impact.
>
>
> A: We thank the reviewer for their valuable suggestion. Following this, we conducted an experiment to assess the robustness of the proposed method against input distortion. This involved adding Gaussian noise to the input frames of Atari games. Specifically, we set the mean to 0 and experimented with various standard deviation values. The results are detailed in the table below:
>
> |  | Alien | MsPacman | SpaceInvaders | StarGunner |
> | --- | --- | --- | --- | --- |
> | MDT | 3.8% | 13.2% | 8.6% | 2.3% |
> | DT-Mem | 51.0% | 69.3% | 53.6% | 62.2% |
> | DT-Mem (std=0.5) | 55.3% | 67.6% | 53.0% | 57.8% |
> | DT-Mem (std=1) | 35.6% | 56.1% | 40.0% | 34.6% |
> | DT-Mem (std=2) | 25.9% | 35.6% | 30.5% | 21.1% |
>
> From the results above, we conclude that the proposed DT-Mem demonstrates greater robustness to noisy inputs compared to the MDT method. This is evident as the DT-Mem consistently outperforms MDT under various levels of Gaussian noise. Notably, the performance with a standard deviation of 0.5 shows minimal difference compared to the no-noise scenario, illustrating DT-Mem's effectiveness in mitigating the impact of varying input distortions.

---

### Author Response · Authors · 2023-11-14
**Summary of the paper revision**

## Summary of the paper revision

1. **Color-Coded Revisions:** The paper has been updated based on the valuable feedback from each reviewer. We've used color coding in the revised document for easy identification of changes: blue for Reviewer 1, green for Reviewer 2, orange for Reviewer 3, and yellow for Reviewer 4.
2. **Figure 2 Update:** Revised Figure 2 to more effectively illustrate the memory module operations.
3. **Terminology Clarification:** Terms have been clarified as per the reviewers' suggestions.
4. **LoRA vs. Full Fine-Tuning:** Conducted additional experiments to compare LoRA with full fine-tuning of the memory module.
5. **Input Misleading Analysis:** Investigated input misleading issues to demonstrate the robustness of the memory module.
6. **Offline RL Section:** Introduced a new section on offline reinforcement learning in the revised paper.
7. **Expanded Related Works:** Added more references to related papers in the related works section.
8. **Detailed Algorithm in Appendix:** Included a detailed algorithm of memory operations in the Appendix.
9. **Typo Corrections:** Addressed and corrected typos throughout the paper.

---

> ### Author Response · Authors · 2023-11-20
>
> Dear Reviewers,
>
> Thank you for your time. We know it’s a busy moment, but we would like to know if there’s anything else we can do to clarify your concerns about our paper.
>
> Please see the summary of the paper revision above.
>
> ***Reviewers gRQQ, hmxu:***
> Thank you for your positive feedback, we hope the rebuttal helped in answering your questions. Please let us know if you have other comments and concerns.
>
> ***Reviewer qzjQ:***
> Your main concern was about the clarity of our approach, its underlying motivation, and its implementation details, particularly regarding the memory initialization process and the application of the LoRA technique.
>
> In the response, we provided detailed explanations to clarify these points. We elaborated on the conceptual framework connecting memorization and forgetting in neural networks with generalization performance, and how DT-Mem is designed to improve this. We also clarified the single initialization process of the memory module during training, which is a distinctive aspect of our approach.
>
> We also implemented your proposed Memory Maze benchmark to evaluate the performance of DT-Mem in long-term dependency tasks. The results from this benchmark were included to demonstrate the superior performance of DT-Mem in handling complex tasks, thereby addressing your concerns about its effectiveness and generalizability.
>
> Is there anything else that we can do to address your concerns?
>
> ***Reviewer NgFj:***
> Your main concern was about the novelty of our approach, specifically regarding the integration of an external memory into Transformers, the comparison with other memory-augmented Transformer models, and the need for more detailed background content and related work in our paper.
>
> In the response, we highlighted the unique aspects of our working memory module inspired by human cognition, specifically designed for decision-making agents. We emphasized the novel application of LoRA in conjunction with our memory module in LLM-based decision-making agents. We also provided a detailed comparison of our memory read and write mechanism with other models like ∞-former, LONGMEM, and KNN-transformer, and addressed the lack of sufficient background content by expanding on Decision Transformers and offline RL settings in our revised paper.
>
> We also expanded related work section to cover more memory-based Transformer papers. We conducted additional experiments and ablation studies as requested, such as the comparison of Full Fine-Tuning (FFT) with LoRA using Atari games and provided visualizations to demonstrate how our model effectively uses memory in decision-making.
>
> Is there anything else that we can do to address your concerns?
>
> Thanks!

---

### Meta-Review · Area_Chair_HGCr · 2023-12-13

**Metareview:**

This paper presents a memory module for Transformer-based architecture, which introduces an internal memory mechanism into the Reinforcement Learning (RL) field and improves training efficiency and generalization. The proposed memory retrieval accesses the memory matrix based on content-based addressing. The memory module is integrated with a pre-trained Decision Transformer (GPT2 architecture) for multi-task RL settings, coupled with a low-rank adaptation fine-tuning method (LoRA). The effectiveness of this approach is evaluated on multi-game Atari and meta-world object manipulation benchmarks, demonstrating consistent improvements in terms of generalization, adaptation, and scaling.

The general motivation behind the paper is well-founded, as a memory module can enhance memorization and be potentially beneficial for the multi-task RL setting. The experimental results show good performance with clear improvement gains.

However, the presentation of the paper needs improvement, including a clearer description of the approach, its motivation, and implementation details. Furthermore, the current baselines only include RMDT as a memory-based Transformer, which is not sufficient. Additional baselines should be included in the experiments to make the results more convincing.

**Justification For Why Not Higher Score:**

The presentation of the paper needs improvement, including a clearer description of the approach, its motivation, and implementation details. Furthermore, the current baselines only include RMDT as a memory-based Transformer, which is not sufficient. Additional baselines should be included in the experiments to make the results more convincing.

**Justification For Why Not Lower Score:**

N/A

---

### Decision · Program_Chairs · 2024-01-16

Reject